

# 1 The AICC2023 chronological framework and associated timescale

# 2 for the EPICA Dome C ice core.

**Marie Bouchet[1],** Amaëlle Landais[1], Antoine Grisart[1], Frédéric Parrenin[2], Frédéric Prié[1],
Roxanne Jacob[1], Elise Fourré[1], Emilie Capron[2], Dominique Raynaud[2], Vladimir Ya Lipenkov[3],
Marie-France Loutre[4,5], Thomas Extier[6], Anders Svensson[7], Etienne Legrain[2], Patricia
Martinerie[2], Markus Leuenberger[8], Wei Jiang[9], Florian Ritterbusch[9], Zheng-Tian Lu[9], Guo-Min
Yang[9].
[1]Laboratoire des Sciences du Climat et de l'Environnement, UMR8212, CNRS, 91190 Gif sur Yvette, France.
[2]Univ. Grenoble Alpes, CNRS, INRAE, IRD, Grenoble INP, IGE, 38000 Grenoble, France
[3]Arctic and Antarctic Research Institute, 199397 St. Petersburg, Russia.
[4]PAGES International Project Office University of Bern, 3012 Bern, Switzerland.
[5]Université catholique de Louvain, B-1348 Louvain-la-Neuve, Belgium.
[6]Univ. Bordeaux, CNRS, Bordeaux INP, EPOC, UMR 5805, 33600 Pessac, France.
[7]Niels Bohr Institute, University of Copenhagen, 2100 Copenhagen, Denmark.
[8]Physics Institute, University of Bern, 3012 Bern, Switzerland.
[9]Hefei National Laboratory, University of Science and Technology of China, Hefei, 230026, China.
*Correspondance to:* Marie Bouchet (marie.bouchet@lsce.ipsl.fr)
**Abstract.** The EPICA (European Project for Ice Coring in Antarctica) Dome C (EDC) ice core drilling in East
Antarctica reaches a depth of 3260 m. The reference EDC chronology (AICC2012) provides an age vs depth
relationship covering the last 800 kyr (thousands of years) with an absolute uncertainty rising up to 8,000 years at
the bottom of the ice core. The origins of this relatively large uncertainty are threefold: (1) the $\delta^{18}O_{atm}, \delta O_2/N_2$
and total air content (TAC) records are poorly resolved and discontinuous over the last 800 kyr, (2) the three orbital
tools are not used simultaneously and (3) large uncertainties are associated with their orbital targets. Here, we
present new highly resolved $\delta^{18}O_{atm}, \delta O_2/N_2$ and $\delta^{15}N$ measurements for EDC ice core covering the last five
glacial - interglacial transitions as well as novel absolute $^{81}Kr$ ages. We have compiled chronological and
glaciological information including novel orbital age markers from new data on EDC ice core as well as accurate
firn modeling estimates in a Bayesian dating tool to construct the new AICC2023 chronology. The average
uncertainty of the ice chronology is reduced from 2,500 years to 1,800 years in AICC2023 over the last 800 kyr.
The new timescale diverges from AICC2012 and suggests age shifts reaching 3,800 years towards older ages over
Marine Isotopes Stages (MIS) 5, 11 and 19. But, the coherency between the new AICC2023 timescale and
independent chronologies of other archives (Italian Lacustrine succession from Sulmona Basin, Dome Fuji ice
core and northern Alpine speleothems) is improved by 1,000 to 2,000 years over these time intervals.

## 36 1 Introduction

Deep polar ice cores are unique archives of past climate and their investigation is valuable to study mechanisms
governing the Earth's climate variations. Precise chronologies are key to identify the successions and lengths of



climatic events, along with exploring phase relationships between the external forcing (changes in the Earth's
orbit) and the diverse climatic responses (variations in temperature and atmospheric greenhouse gas
concentrations). To date ice cores, we need to construct two separate chronologies: one for the ice and one for the
younger air trapped in bubbles. Due to the thinning of ice horizontal layers as we go down in depth, a wide timespan
of paleoclimatic information is zipped within the deepest part of the ice sheet. Therefore, many of the ice core
community ongoing efforts focus on improving deep ice cores timescales for ice and gas phases, as well as
extending them further back in time (Crotti et al., 2021; Oyabu et al., 2022). Ice cores drilled at sites characterized
by a high accumulation rate of snow at surface (10 to 30 cm/year) can be dated by counting ice layers deposited
year after year (Svensson et al., 2008; Sigl et al., 2016). On the contrary, East Antarctica sites are associated with
very low accumulation rates (1 to 5 cm/year) which prevent annual layers from being identified and counted. As
a consequence, chronologies of ice cores at low-accumulation sites are commonly established using ice flow and
accumulation models tied up with chronological and glaciological constraints (Veres et al., 2013; Bazin et al.,
2013; Parrenin et al., 2017).

Glaciological modeling has been historically used to date Greenlandic and Antarctic ice cores. A

unidimensional ice flow model was first applied to the Camp Century ice core (Dansgaard and Johnsen, 1969),
and later to other ice cores such as the ones drilled at EDC and Dome Fuji (EPICA members, 2004; Parrenin et
al., 2007). First, water isotopes ($\delta$D or $\delta^{18}$O) measurements provide estimates of past evolution of the accumulation
rate of snow and temperature at surface. Then, an ice flow model (Parrenin et al., 2004) takes as inputs past
accumulation together with a vertical velocity depth-profile through the ice sheet to determine the thinning of
annual snow/ice layers in time, and therefore the ice timescale. This approach is very sensitive to some poorly
known parameters such as boundary conditions. For this reason, the glaciological modeling approach is
complemented with chronological constraints (gas or ice age known at certain depth levels).

Chronological constraints obtained either by measurement of radionuclides or by synchronization to a

curve of reference are established for both ice and gas timescales. For building long chronologies, some time
constraints can be obtained from the $^{10}$Be series measured in ice. The $^{10}$Be cosmogenic nuclide is produced at
different rate depending on the solar activity and its arrival on Earth is modulated by the strength of the Earth's
magnetic field (Yiou et al., 1997; Raisbeck et al., 2007; Heaton et al., 2021). Some links hence exist between $^{10}$Be
flux and precisely dated magnetic events such as the Laschamp excursion, an abrupt decline in the geomagnetic
field magnitude occurring at about 41 ka BP (thousand years before 1950) and visible as a positive excursion in
the $^{10}$Be flux records in ice cores (Raisbeck et al., 2017; Lascu et al., 2016). $^{40}$Ar measurements in the gas phase
of Antarctic ice cores also provide dating constraints for old ice, especially for non-continuous stratigraphic
sequences (Yan et al., 2019). $^{40}$Ar is produced in solid earth by the radioactive decay of $^{40}$K leading to an increasing
concentration of $^{40}$Ar in the atmosphere at a rate of $0.066 \pm 0.006$‰ Myr$^{-1}$ (Bender et al., 2008). Recently, the
possibility of measuring $^{81}$Kr in ice samples of a few kg gave a new absolute dating tool for ice cores. $^{81}$Kr is a
radioactive isotope that is suitable for dating ice cores in the range from 0.03 to 1.3 Ma BP (million years before
1950), making it perfectly adapted for Antarctic ice core dating (Crotti et al., 2021; Buizert et al., 2014).

To further constrain oldest ice core chronologies, the so-called "orbital dating" tools are also used. These

tools consist in aligning some tracers measured in ice cores to the Earth orbital series, called targets, whose
fluctuations in time are accurately calculated from the known variations of orbital parameters (Berger, 1978;
Laskar et al., 2011). The synchronization of the orbital tracer with its target provides ice or gas age constraints. So





far, three orbital dating tools have been developed: $\delta^{18}O$ of $O_2$ ($\delta^{18}O_{atm}$), $\delta O_2/N_2$ and total air content (TAC).
$\delta^{18}O_{atm}$ was typically aligned with the precession parameter (or with the $21^{st}$ June insolation at 65°North) delayed
by 5,000 years because such a lag between $\delta^{18}O_{atm}$ and its orbital target was observed during the last deglaciation
(Dreyfus et al., 2007; Shackleton, 2000). However, variations in the phasing between $\delta^{18}O_{atm}$ and precession have
been suspected (Jouzel et al., 2002) and identified since (Bazin et al., 2016), and millennial-scale events (as
Heinrich-like events) occurring during deglaciations have been shown to delay the response of $\delta^{18}O_{atm}$ to orbital
forcing (Extier et al., 2018). Because there was a significant unpredictability in the lag between $\delta^{18}O_{atm}$ and its
orbital target, a large uncertainty in the $\delta^{18}O_{atm}$ based tie points (up to 6,000 years) was assigned in the
construction of AICC2012 (Bazin et al., 2013). To improve the accuracy of the gas timescale, Extier et al. (2018)
rather aligned the variations of $\delta^{18}O_{atm}$ to the $\delta^{18}O_{calcite}$ recorded in absolute dated East Asian speleothems
between 640 and 100 ka BP. Indeed, the two records show similar orbital (related to the $21^{st}$ July insolation at 65°
North) and millennial variabilities, which may correspond to southward shifts in the InterTropical Convergence
Zone (ITCZ) position, themselves linked to Heinrich-like events as supported by the modeling study of Reutenauer
et al. (2015).

In parallel, Bender (2002) observed that the elemental ratio $\delta O_2/N_2$ of air trapped in Vostok ice core

appears to vary in phase with the $21^{st}$ of December insolation at 78° South (Vostok latitude) between 400 and 160
ka BP. Subsequent observations led Bender (2002) to assert that local summer solstice insolation affects near-
surface snow metamorphism and that this imprint is preserved as snow densifies in the firn and, later on, affects
the ratio $\delta O_2/N_2$ measured in air bubbles formed at the lock-in-zone. Wiggle matching between $\delta O_2/N_2$ and local
summer solstice insolation has been used to construct orbital timescales for Dome Fuji, Vostok and EDC ice cores
reaching back 360, 400 and 800 ka BP respectively, with a chronological uncertainty for each $\delta O_2/N_2$ tie point
estimated between 800 and 4,000 years (Kawamura et al., 2007; Suwa & Bender, 2008; Bazin et al., 2013).
Finally, Raynaud et al. (2007) found very similar spectral properties between the TAC record of EDC and the
integrated summer insolation at 75°South (ISI) obtained by a summation over a year of all daily local insolation
above a certain threshold over the last 440 kyr. As for $\delta O_2/N_2$, these similarities may be explained by the insolation
imprint in near-surface snow well preserved down to the lock-in zone, where it could affect the air content in deep
ice although the physical mechanisms involved during the snow and firn densification for $\delta O_2/N_2$ and TAC are
likely different (Lipenkov et al., 2011). Lately, Bazin et al. (2013) made use of TAC to constrain Vostok and EDC
ice core chronologies back to 430 ka BP with an uncertainty for each TAC tie point varying between 3,000 and
7,000 years. Although these three orbital tools complement each other (TAC and $\delta O_2/N_2$ inferred ages agree
within less than 1,000 years between 390 and 160 ka BP for the Vostok ice core, Lipenkov et al., 2011), they
hardly ever have been employed together. Plus, they are often associated with large uncertainties (reaching 7,000
years) which lie in the choice of the appropriate orbital target, in its alignment with ice core records that can be
ambiguous during periods of low eccentricity in the Earth's orbit (leading to low-amplitude insolation variations)
and in the poor quality of the signals measured in the deepest section of the cores.

To connect ice and gas timescales, the estimation of the lock-in-depth (LID), indicating the lowest depth where

air is trapped in enclosed bubbles and no longer diffuses (Buizert et al., 2013), is used to calculate the ice/gas age
difference. Measurements of $\delta^{15}N$ from $N_2$ yield a first estimate of this depth and the LID can also be calculated
with firn densification modeling (Bréant et al., 2017; Goujon et al., 2003).





For many years, each polar ice core was characterized by its peculiar timescale which was not naturally
consistent with other ice cores timescales. To address this issue, other measurements provide relative dating
constraints (stratigraphic links) improving the coherency between timescales of ice cores from both hemispheres.
The synchronization of globally well-mixed atmospheric methane ice core records gives tie points with an accuracy
of a few decades to several centuries (60-1,500 years) (Lemieux-Dudon et al., 2010; Epifanio et al., 2020). Climate
independent events, such as large volcanic eruptions, can be observed in ice cores from Greenland and Antarctica
via singular patterns of the distribution of sulfate. Identification of these deposits permits to precisely synchronize
several ice cores (within 5 to 150 years) (Svensson et al., 2020).
In order to integrate stratigraphic matching, independent synchronization and absolute dating constraints as
well as glaciological modeling to produce coherent ice core chronologies, researchers developed Bayesian dating
tools such as Datice (Lemieux-Dudon et al., 2010), IceChrono1 (Parrenin et al., 2015) and Paleochrono (Parrenin
et al., 2021). These tools use an inverse method combining all chronological information to provide a coherent age
scale for several ice cores. These probabilistic tools adjust prior estimates of ice and gas chronologies built with a
glaciological model (background scenario) so that they respect chronological constraints.
Here we focus on the chronology of the EDC deep ice core. The EPICA project provided two cores in
East Antarctica including one at Dome C (EDC, 2004). The second (and final) drilling attempt at Dome C gave
the 3260 m long EDC99 core, whose drilling has been willingly stopped at 15 m above bedrock due to expected
presence of melt water. EDC furnishes the oldest continuous ice core record so far, covering the last 800 kyr
(EPICA community members, 2004; Jouzel et al., 2007).
Bazin et al. (2013) and Veres et al. (2013) used the probabilistic dating tool Datice to establish the
coherent chronology AICC2012 back to 800 ka BP for five ice cores including EDC, Vostok, EPICA Dronning
Maud Land ice core (EDML), North Greenland Ice core Project (NGRIP) and Talos Dome Ice core (TALDICE).
To determine EDC age scale, they used various orbital dating constraints including: 39 tie points attached to a
6,000 years uncertainty derived from $\delta^{18}O_{atm}$ tuning to 5,000 years delayed precession between 800 and 300 ka
BP, 20 tie points associated with a 4,000 years uncertainty from $\delta O_2/N_2$ alignment to local summer solstice
insolation between 800 and 300 ka BP, and 14 tie points linked to an uncertainty between 3,000 and 7,000 years
using TAC synchronised to integrated summer insolation between 430 and 0 ka BP. However, due to the lack of
data for the orbital dating approach, AICC2012 uncertainty is of 2,500 years on average, reaching 8,000 years at
the bottom of the core. The origins of AICC2012 uncertainty can be divided in the following points: (i) discrepancy
between $\delta^{18}O_{atm}$, $\delta O_2/N_2$ and TAC series and their orbital target; (ii) discontinuity and poor quality of the $\delta O_2/N_2$
and TAC records; (iii) uncertainty on the phasing between $\delta^{18}O_{atm}$ and precession; (iv) poor constraint on the LID
scenario due to a disagreement between $\delta^{15}N$ data and firn modeling estimates (Bréant et al., 2017).
It is now possible to address each source of uncertainty thanks to recent advances: (i) Since AICC2012,
the $\delta^{18}O_{atm}$, $\delta O_2/N_2$ and TAC records have been extended, now covering the last 800 kyr (Extier et al., 2018c,
b). In addition, new highly resolved $\delta^{18}O_{atm}$ and $\delta O_2/N_2$ measurements are available over several glacial
terminations (TII, III, IV, V and VI) (Grisart, 2023). (ii) Extier et al. (2018) recently suggested a $\delta^{18}O_{atm}$ based
timescale using $\delta^{18}O_{calcite}$ of East Asian speleothems as an alternative tuning target to precession. This choice
reduces the chronological uncertainty between 640 and 100 ka BP. (iii) Finally, new highly resolved $\delta^{15}N$ data
covering the Terminations II to VI are available (Grisart, 2023). In parallel, firn densification models have been





progressively improved and the model described in Bréant et al. (2017) can be employed to estimate LID evolution
in the past when $\delta^{15}N$ data are still missing.
In this work, we implement new absolute age constraints spanning the last 800 kyr derived from $^{81}Kr$ measured
in air trapped in EDC ice core as well as new orbital age constraints obtained by synchronizing up-to-date EDC
records with their orbital target. We combine these data to recent volcanic matching and methane records
synchronization which provide additional stratigraphic links, relating EDC to other ice cores over the past 122 kyr
(Svensson et al., 2020; Baumgartner et al., 2014). Finally, we propose the new chronology AICC2023 with reduced
chronological uncertainties.

## 2 Methods

### 2.1 Dating strategy

The Paleochrono Python software is a probabilistic dating tool similar to Datice and Icechrono1 with
improved mathematical, numerical and programming capacities (Parrenin et al., 2021). The dating strategy of
Paleochrono relies on the Bayesian inference of three glaciological functions forming the input background
scenario: accumulation rate ($A$), thinning of annual ice layers ($\tau$) and Lock-In-Depth ($LID$). The three variables
evolve along the ice core depth $z$ and are used to estimate the ice ($\psi$) and gas ($\chi$) age profiles as follows:

$$\psi(z) = \int_{0}^{z} \frac{D(z')}{\tau(z')A(z')}dz' \qquad (1)$$

$$\chi(z) = \psi\big(z - \Delta depth(z)\big) \qquad (2)$$

$$\int_{z-\Delta depth(z)}^{z} \frac{D(z')}{\tau(z')}dz' = LID(z) \times \frac{D}{\tau}\bigg|_{firn}^{0} \qquad (3)$$

where $D$ is the relative density of the snow/ice and $\frac{D}{\tau}\big|_{firn}^{0}$ the average value of $\frac{D}{\tau}$ in the firn when the air particle
was at the lock-in-depth (this parameter is usually ~0.7, Parrenin et al., 2012). The age scales are further
constrained to respect chronological constraints identified from observations. To specify the credibility of the
background scenario for the age scales and the chronological constraints, the glaciological functions
(accumulation, thinning and LID) and the chronological information can be mathematically expressed as
probability densities which are presumed to be Gaussian and independent (i.e. decorrelated between them). Thus,
the inference is based on the Least Square optimisation method (implying all probability densities Gaussian). It is
numerically solved using the Trust Region algorithm (assuming that the model is roughly linear around the
solution) and the Jacobian of the model is evaluated analytically for an improved computation time. As a result,
the best adjustment between the background scenario and chronological observations is found, providing the most
probable scenario as a posterior evaluation of the three glaciological functions and hence chronologies for ice and
air. For each ice core, the input files for Paleochrono are the following: (i) the background values of the three
glaciological functions with depth, (ii) gas and ice stratigraphic links, (iii) gas and ice dated horizons, which are
tie points derived for one core from absolute and synchronization dating methods, (iv) gas and ice intervals of



known durations and (v) depth difference estimates between the same event recorded in the gas and ice matrix
(Δdepth). Specific relative or absolute uncertainties are attached to each of these parameters in each input file.

In this study, we added numerous gas and ice dated horizons for EDC as well as an updated background
scenario for the LID. Then, to construct a new chronology for EDC ice core that is consistent with the timescales
of Vostok, TALDICE, EDML and NGRIP ice cores, we followed the same strategy as for the construction of
AICC2012. Glaciological background parameters and dating constraints for Vostok, TALDICE, EDML, NGRIP
and EDC drillings are compiled in one run of Paleochrono to obtain AICC2023. Vostok, TALDICE, EDML and
NGRIP background parameters and dating constraints are extracted from Bazin et al. (2013) except for: (i) new
Vostok gas age constraints determined from the alignment of $\delta^{18}O_{atm}$ and East Asian $\delta^{18}O_{calcite}$ records as for
EDC (see supplementary Fig. S5), (ii) new TALDICE background parameters and (iii) age constraints from Crotti
et al. (2021) and (iv) corrected LID background scenarios for Vostok and EDML sites (see supplementary Fig.
S6). In order to prevent any confusion with reference ice core timescales, the new AICC2023 chronology for
NGRIP is compelled to respect exactly the layer-counted GICC05 timescale over the last 60 kyr (Andersen et al.,
2006). For this reason, we did not use the methodology described by Lemieux-Dudon et al. (2015) which
implemented layer counting as a constraint on the duration of events in the dating tool, inducing a slight shift
(maximum 410 years) on the AICC2012 timescale. The resulting Paleochrono experiment provides the new
official chronology AICC2023 for the EDC ice core. The contingent timescales obtained for the four other sites
are not the subject of this study but are also provided in the Supplementary Material.

**2.2 Analytical method**

**2.2.1    $\delta^{18}O_{atm}$, $\delta O_2/N_2$ and $\delta^{15}N$**

The measurements of the isotopic and elemental compositions of $O_2$ and $N_2$ were performed by Grisart
(2023) at LSCE following the method described by Bréant et al. (2019) and Extier et al. (2018). The air trapped in
the EDC ice core is extracted using the semi-automatic line which eliminates $CO_2$ and $H_2O$. 30 to 40 g samples
are prepared in a cold environment (-20°C), their exterior layer (3-5mm) is removed so that there is no exchange
with atmospheric air and each sample is cut in two replicates. Each day, three ice samples (and replicates) are
placed in six flasks and the atmospheric air is evacuated from the flasks. Samples are then melted and left at
ambient temperature for approximately 1h30 in order to extract the air trapped in ice samples. The extracted air is
then cryogenically trapped within a dedicated manifold immersed in liquid helium (Bazin et al., 2016). Along the
way to the cryogenic trap, the air goes through cold traps to remove $H_2O$ and $CO_2$. Two additional samples
containing exterior modern air are processed through the same line every day for calibration and for monitoring
the analytical set-up. Lastly, the $\delta^{15}N$, $\delta^{18}O$ of $O_2$ and $\delta O_2/N_2$ of each sample are measured by a dual inlet Delta
V plus (Thermo Electron Corporation) mass spectrometer.

Classical corrections are applied on the measurements (pressure imbalance, chemical slopes, as per
Landais et al., 2003). In addition, $\delta^{15}N$ data are used to get the values of atmospheric $\delta^{18}O$ of $O_2$ and $\delta O_2/N_2$ after
gravitational fractionation occurred in the firn, so that $\delta^{18}O_{atm} = \delta^{18}O$ of $O_2 - 2 \times \delta^{15}N$ and $\delta O_2/N_{2(corr)} =$
$\delta O_2/N_{2(raw)} - 4 \times \delta^{15}N$ (Landais et al., 2003; Bazin et al., 2016; Extier et al., 2018). Note that our samples were
stored at -50°C since drilling so that no correction for gas loss was applied (see Supplementary Material).





Existing and new EDC data are compiled in Table 1. The resulting data set pooled standard deviations
for the new measurements are of 0.006, 0.03 and 0.04 ‰ for $\delta^{15}N$, $\delta^{18}O_{atm}$ and $\delta O_2/N_2$ respectively.
**Table 1**. **Information on isotopic and elemental compositions measured in air trapped in EDC ice core**. *Details on
storage and measurement conditions of $\delta O_2/N_2$ are available in the Supplementary Material.

| | $\delta^{18}O_{atm}$ | | | $\delta O_2/N_2$* | | | $\delta^{15}N$ | | |
|---|---|---|---|---|---|---|---|---|---|
| | Depth (m) | AICC2012 gas age (ka BP) | Resolution (kyr) | Depth (m) | AICC2012 ice age (ka BP) | Resolution (kyr) | Depth (m) | AICC2012 gas age (ka BP) | Resolution (kyr) |
| **AICC2012** | | | | | | | | | |
| Bazin et al. (2013); Dreyfus et al. (2007, 2008, 2010); Landais et al. (2012) | 2479 - 3260 | 300 - 800 | 1 - 1.5 | 2480 - 3260 | 300 - 800 | 2.5 | 346 - 578 | 11 - 27 | 0.35 - 0.38 |
| | | | | | | | 1090 - 1169 | 75 - 83 | 1.4 |
| | | | | | | | 1389 - 3260 | 100 - 800 | 2.4 |
| Bazin et al. (2016) | 1300 - 1903 | 90 - 160 | 1.1 | 1300 - 1903 | 93 - 163 | 2.37 | | | |
| | 2657 - 3260 | 370 - 800 | | 2595 - 3260 | 340 - 800 | 2.08 | | | |
| Extier et al. (2018b, 2018c) | 1872 - 2665 | 153 - 374 | 0.16 - 0.7 | 1904 - 2562 | 164 - 332 | 2 - 2.5 | | | |
| Bréant et al. (2019) | | | | | | | 1904 - 2580 | 160.2 - 334.5 | 1.013 |
| This work (Grisart, 2023) | 1489.95 - 1832.6 | 108.0 - 136.3 | 0.333 | 1489.95 - 1832.6 | 111.4 - 148.9 | 0.441 | 1489.95 - 1832.6 | 108.0 - 136.3 | 0.333 |
| | 1995.95 - 2350.15 | 180.6 - 255.8 | 0.437 | 1995.95 - 2350.15 | 183.9 - 259.6 | 0.437 | 1995.95 - 2350.15 | 180.6 - 255.8 | 0.437 |
| | 2555.85 - 2633.4 | 328.3 - 346.8 | 0.356 | 2555.85 - 2633.4 | 330.5 - 360.6 | 0.579 | 2555.85 - 2633.4 | 328.3 - 346.8 | 0.356 |
| | 2744.5 - 2797.85 | 408.7 - 445.9 | 0.744 | 2744.5 - 2797.85 | 410.7 - 449.6 | 0.779 | 2744.5 - 2797.85 | 408.7 - 445.9 | 0.744 |
| | 2873.75 - 2910.6 | 508.1 - 535.6 | 1.375 | 2873.75 - 2910.6 | 511.3 - 539.3 | 1.401 | 2873.75 - 2910.6 | 508.1 - 535.6 | 1.375 |


### 2.2.2  Total Air Content

The TAC record has been measured in the entire EDC ice core at the IGE following the barometric method
firstly described by Lipenkov et al. (1995). The TAC record measured in the younger part of the core (400 − 0 ka
BP) has been published in Raynaud et al. (2007) (Table 2). TAC estimates need to be corrected for cut-bubble
effect. After correction, the uncertainty in TAC values is of about 1% and the analysis replicability is better than

236 1%.





**Table 2. Information on TAC measurements.**

| | TAC | | |
| --- | --- | --- | --- |
| | Depth (m) | AICC2012 ice age (ka BP) | Resolution (kyr) |
| **AICC2012** (Raynaud et al., 2007) | 115 - 2800 | 0 - 440 | 2.000 |
| *Unpublished* | 2800 - 3260 | 440 - 800 | 2.000 |

### 2.2.3  [81]Kr extraction and analysis

The analytical method is the same as described by Crotti et al. (2021). Three ice samples of 6 kg each are taken from the bottom part of EDC and a slight shaving (1 mm) of the external layer is performed before processing. The air extraction is performed through a manual extraction line following the protocol described in Tian et al. (2019). The ice sample is placed in a 40 L stainless-steel chamber. The atmospheric air is pumped while the chamber is kept at -20°C. The air is then slowly extracted, passing through a water trap, and compressed in a stainless-steel cylinder. The three cylinders are sent to the University of Sciences and Technology of China (USTC, Hefei, China) for Krypton extraction and analysis. Krypton extraction is performed after the methodology of Dong et al. (2019) who set up an automated system for dual separation of Argon and Krypton, composed of a Titanium getter module followed by a Gas-Chromatography separator module. The extracted Krypton is analyzed by the Atom Trap Trace Analysis (ATTA) instrument set up at the Laser Laboratory for Trace Analysis and Precision Measurement (LLTAPM, USTC, Hefei, China), giving the [81]Kr abundance $R_{81}$ in the sample. $R_{81}$ is determined by the number of counted [81]Kr atoms in the sample as compared to the atmospheric reference. The anthropogenic [85]Kr is measured simultaneously with [81]Kr to control any present-day air contamination. Here, the [85]Kr abundance measured in ice samples is inferior to the detection limit, so contamination has occurred.

From the [81]Kr abundance, it is possible to estimate [81]Kr radioactive decay and to calculate the ice samples age. As a noble gas isotope, [81]Kr is globally mixed in the atmosphere and its decay cannot be affected by complex chemical reactions (Lu et al., 2014). [81]Kr half-life ($t_{1/2}$) is estimated to $\simeq 229 \pm 11$ kyr (Baglin, 2008). [81]Kr age can be calculated as per the following equation:

$$\text{age} = -\frac{t_{\frac{1}{2}}}{\ln(2)} \times \ln(R_{81}) \qquad (4)$$

The atmospheric abundance of [81]Kr is not constant in the past and its value is corrected using reconstruction of the geomagnetic field intensity (Zappala et al., 2020). The error in [81]Kr age estimates is estimated from the statistical error of atom counting, from the uncertainty in [81]Kr half-life (inducing a systematic age error) and from the size of the sample (larger sample resulting in a smaller uncertainty).



### 2.3 Firn model

Firn densification models have been progressively improved over the years. While these models generally explain well the evolution of $\delta^{15}N$ in time through changes in the LID, they fail to reproduce values of $\delta^{15}N$ in some regions including coastal areas (Capron et al., 2013) and cold and low accumulation sites such as EDC. This disagreement can be explained by an inaccurate estimate of glacial temperature and accumulation rate at surface (Buizert et al., 2021) and/or by the impossibility to tune empirical firn models to sites with no present-day equivalent in terms of temperature and accumulation rate (Dreyfus et al., 2010; Capron et al., 2013). Lately, the firn model described in Bréant et al. (2017) has been developed from the IGE (Institute of Environmental Geosciences) model (Barnola et al., 1991; Pimienta & Duval, 1987; Arnaud et al., 2000; Goujon et al., 2003) by implementing a dependency of the firn densification rate on temperature and impurities. The temperature dependence is added on the classical formulation of the densification rate following an Arrhenius law with an activation energy Q as per $\exp(-Q/RT)$ with R the perfect gas constant and T the firn temperature. Rather than using a constant activation energy (Goujon et al., 2003), Bréant et al. (2017) stated that the value of the activation energy should be contingent on the firn temperature value as observed in material science where the temperature dependency exhibits the predominance of one physical mechanism among others for a material compaction at specific temperature. Through several sensitivity tests, Bréant et al. (2017) adjusted three values for activation energy on three different temperature ranges to reproduce best the $\delta^{15}N$ evolution over the last deglaciation at East Antarctic sites. The firn model also considers that firn densification is facilitated by the dissolution of impurities within the snow (Freitag et al., 2013). If the impurity content in snow (i.e. concentration of calcium ions) is superior to a certain threshold, the densification rate dependency to impurities is traduced by a relationship between the new activation energy $Q'$ and the concentration of calcium ions $[Ca^{2+}]$: $Q' = f_1 \times (1 - \beta \ln\left(\frac{[Ca^{2+}]}{[Ca^{2+}]_{threshold}}\right)) \times Q$ (Freitag et al., 2013). Bréant et al. (2017) assumed the impurity effect equal for all physical mechanisms and tuned $\beta$ and $f_1$ constants so that the modelled - $\delta^{15}N$ data mismatch is minimized over the last glacial termination at cold East Antarctic sites.

As a consequence, and in addition to our new extensive $\delta^{15}N$ dataset, we have chosen to use here the firn model approach of Bréant et al. (2017). In order to make a correct calculation of uncertainties linked to firn modeling at EDC, we ran two tests of the model with and without including the impurity concentration parameter (Table 3).

**Table 3. Information about the two runs of the model.**

| Parameter | Test 1 | Test 2 |
|---|---|---|
| **Activation energy Q (densification rate)** | Q depending on temperature (Bréant et al., 2017) | Q depending on temperature (Bréant et al., 2017) |
| **Impurity inclusion** | Yes | No |

The firn densification model takes as input scenarios of temperature and accumulation rate at the surface. It computes both the LID and the thermal gradient in the firn ($\Delta T$), and then deduces the $\delta^{15}N_{therm} = \Omega \cdot \Delta T$ with $\Omega$



the thermal fractionation coefficient (Grachev and Severinghaus, 2003). The final $\delta^{15}N$ is calculated as $\delta^{15}N =$
$\delta^{15}N_{grav} + \delta^{15}N_{therm}$ and $\delta^{15}N_{grav} \simeq LID \cdot \frac{g}{RT}$ (first order approximation) with $g$ the gravitational acceleration
$(9.8 \text{ ms}^{-2})$, $R$ the gas constant $(8.314 \text{ Jmol}^{-1}\text{K}^{-1})$ and $T$ the mean firn temperature (K).
**3 Results**
**3.1 $^{81}$Kr age constraints**
Three ice samples from the bottom part of EDC have been analyzed and provide three age estimates displayed
in Table 4: 629, 788 and 887 ka BP with statistical age uncertainties between 30 and 50 kyr, and a 4.8 % systematic
error due to the uncertainty in the half-life of $^{81}$Kr. The deepest sample suggests the presence of ice older than 800
ka BP below the 3200 m depth level and further dating studies would be valuable in exploring whether the
stratigraphy of EDC lowermost section is continuous, although this is beyond the scope of this work.
**Table 4. Ice samples details and radio krypton dating results**. Reported errors are 1-σ errors. Upper limits have a 90%
confidence level. The average $^{85}$Kr activity in the northern hemisphere is about 75 dpm/cc in 2017. The measured $^{85}$Kr
concentrations are inferior to the detection limit, verifying that no relevant contamination with modern air has occurred. In
addition to the statistical error on the $^{81}$Kr age from atom counting, a systematic error due to the uncertainty in the half-life of
$^{81}$Kr is considered. This error would shift the calculated $^{81}$Kr ages up or down for all ice samples. [a]dpm/cc = decays per minute
per cubic centimeter STP of krypton (conversion: 100 dpm/cc corresponds to $^{85}$Kr/Kr = $3.03 \times 10^{11}$). [b]pMKr = percent Modern
Krypton (Jiang et al., 2023).

| Depth (m) | Air extracted / Ice weight (mL/kg) | Sample Used (μL STP, Kr) | Analysis Date | $^{85}$Kr (dpm/cc)[a] | $^{81}$Kr (pMKr)[b] | $^{81}$Kr − age (ka BP) age$^{+stat+sys}_{-stat-sys}$ |
|---|---|---|---|---|---|---|
| 3013-3024 | 440/6.0 | ~0.46 | 18 Dec 2019 | < 0.77 | $15.1^{+1.4}_{-1.4}$ | $629^{+34+31}_{-29-31}$ |
| 3144-3161 | 600/8.4 | ~0.67 | 30 Dec 2019 | < 0.67 | $9.6^{+1.0}_{-1.0}$ | $788^{+36+38}_{-33-38}$ |
| 3216-3225 | 415/6.4 | ~0.43 | 16 Jan 2020 | < 1.17 | $7.1^{+1.0}_{-1.0}$ | $887^{+51+43}_{-44-43}$ |


**3.2 Determination of orbital age constraints using new data**
**3.2.1 $\delta O_2/N_2$**
In this work, new highly resolved $\delta O_2/N_2$ data on EDC ice core are presented over Terminations II, III, IV,
V and VI (Fig. 1). As these novel $\delta O_2/N_2$ measurements have been performed on ice samples stored at -50℃,
there is no storage effect and they can directly be merged with the 800 kyr long record of Extier et al. (2018c)
(Table 1). The new dataset improves the resolution of the long EDC record, reaching sub-millennial scale accuracy
over MIS 5, 7, 9 and in particular over MIS 11 and MIS 13, periods of sparsity in the ancient record (Extier et al.,
2018c). Although the two datasets agree well over recent periods (last 350 kyr), they show some discrepancies
during older periods (between 550 and 375 ka BP, see Fig. 1). Such dissimilarities are observed over MIS 11
(between 424 and 374 ka BP) where the sampling resolution of the previous dataset is particularly low (2,500
years). In addition, the MIS 11 is a period characterised by a low eccentricity in the Earth orbit, inducing subdued
variations of insolation, causing $\delta O_2/N_2$ changes of smaller magnitude and leading to lower signal to noise ratio.
Data by Landais et al. (2012) (shown by purple squares on Fig. 1 and S2) are consistent with the highly resolved





data presented here, supporting the relevance of the new dataset over this period. Over Termination VI (from 550
to 510 ka BP), the old dataset continuously increases while the novel dataset shows a brief maximum at around
525 ka BP followed by a minimum at around 520 ka BP. These newly revealed variations seem in phase with
insolation variations, suggesting that the new dataset shows an improved agreement with insolation. Still, highly
resolved measurements are needed in the lowermost part of the ice core where noise is significantly altering the
temporal signal.
Following a data processing treatment consistent with the method described in Kawamura et al. (2007), the
compiled dataset is linearly interpolated every 100 years, and then smoothed using a finite-duration impulse
response (FIR) filter with a KaiserBessel20 window (cut-off from 16.7 to 10.0 kyr period, number of coefficients
of 559 for the 800 kyr long record) designed with the software Igor Pro, in order to reject periods inferior to 10,000
years and erase the noise present in the data. Note that using a low-pass (rejecting periods below 15 kyr) or a band-
pass filter (keeping periods between 100 and 15 kyr periods, used by Bazin et al. (2013)) does not alter the peak
positions in the $\delta O_2/N_2$ curve (see supplementary Fig. S1). The noise is particularly significant for highly resolved
$\delta O_2/N_2$ data and without preliminary filtering, it becomes ambiguous to identify the exact peak position (which
needs to be subjectively placed on a 1,000 to 2,000 years interval, see Supplementary Material).
The filter is then applied to the local summer solstice insolation curve to check that it does not induce the shift
of extrema positions by more than 100 years. This condition is verified over the last 800 kyr, except for the peaks
located at the endpoints of the record (respectively around 107 and 788 ka BP) which are then not used for tie
points determination. Outliers in the raw $\delta O_2/N_2$ dataset are discarded if they show an anomaly superior to 3.2 ‰
when compared to the low-pass filtered signal. Five outliers are rejected out of 294 points. The $\delta O_2/N_2$ is
interpolated and filtered again after removal of the outliers.

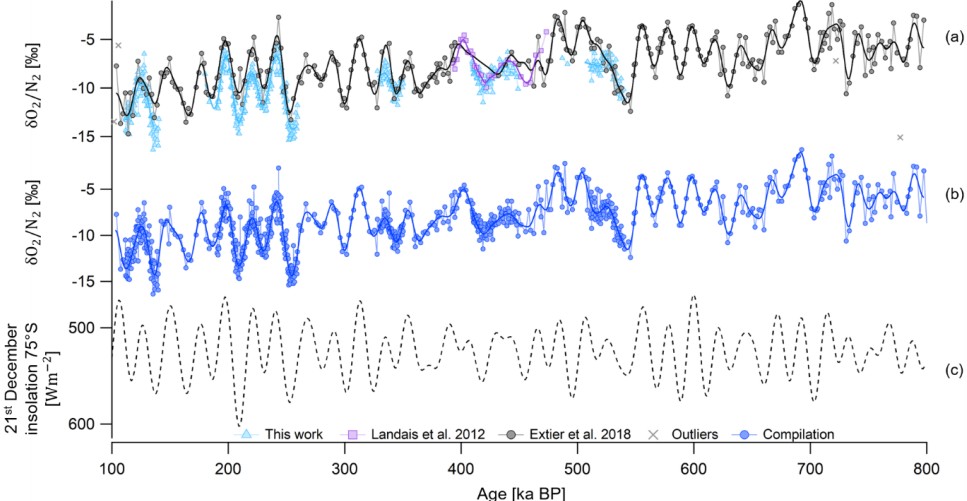

**Figure 1. Evolution of EDC $\delta O_2/N_2$ record between 800 and 100 ka BP.** (a) EDC raw $\delta O_2/N_2$ old data between 800 and
100 ka BP on AICC2012 ice timescale (black circles for data of Extier et al. (2018c) and purple squares for data of Landais et
al. (2012)), outliers (grey crosses) and filtered signal (black and purple lines). EDC raw $\delta O_2/N_2$ new data (blue triangles, this
study) and filtered signals (blue line). (b) Compilation of the two datasets and filtered compiled signal. (c) 21st December



insolation at 75° South on a reversed axis. Zooms between 270 and 100 ka BP and between 570 and 300 ka BP are shown in
supplementary Fig. S1.

The orbital target chosen is the 21st December insolation at 75° South, which is calculated every 100 years
over the last 800 kyr (Laskar et al., 2004). The peak positions in the filtered $\delta O_2/N_2$ compiled signal and in the
summer solstice insolation are detected via an automated method using the zero values of the time derivatives of
the $\delta O_2/N_2$ and its orbital target. Each $\delta O_2/N_2$ maximum is matched to an insolation minimum and each $\delta O_2/N_2$
minimum to an insolation maximum. The data treatment and tie point identification method used here are
consistent with the approach recently performed by Oyabu et al. (2022) on a novel 207 kyr long $\delta O_2/N_2$ record of
DF ice core.
Some periods, such as the MIS 11 (between 450 and 350 ka BP) and older ages (before 600 ka BP), are
characterized by a poor resemblance between the signal and the target. For instance, two or three peaks in the
insolation curve only correspond respectively to one or two peaks in the $\delta O_2/N_2$ data. This could be explained by
a low eccentricity-induced subdued variability in the insolation target and hence in $\delta O_2/N_2$ signal over MIS 11
and to the poor resolution of the $\delta O_2/N_2$ measurements before 600 ka BP. In such cases, the uncertainty associated
with each tie point is ranging from 6 to 10 kyr (precession half period) and some tie points are even discarded (5
points over MIS 11 out of 63 over the last 800 kyr). Otherwise, $\delta O_2/N_2$ seems to evolve in phase with the inverse
summer solstice insolation variations and the tie points uncertainty is set at 3 kyr. This 3 kyr uncertainty was
evaluated by Bazin et al. (2016) after having combined the three $\delta O_2/N_2$ records from Vostok, Dome Fuji and
EDC ice cores. The orbital tuning results in 58 new tie points over the last 800 kyr (displayed in Fig. 2 and compiled
in supplementary Table S1), replacing the 20 tie points used to constrain AICC2012 between 800 and 300 ka BP
that were derived from synchronising mid-slopes of band-pass filtered $\delta O_2/N_2$ with the insolation (Bazin et al.,

2013).



**Figure 2. Alignment of $\delta O_2/N_2$ and insolation between 800 and 100 ka BP**. Extrema in the compiled filtered $\delta O_2/N_2$
dataset (blue plain line, in panel a are identified and matched to extrema in 21st December insolation at 75° S (dash line in panel
b). The matching peaks are linked by black vertical bars. The 0 value in the time derivative of insolation (black line in panel c)
and of the filtered $\delta O_2/N_2$ dataset (blue line, c) corresponds to extreme values in the signals. The determined tie points between
$\delta O_2/N_2$ and insolation are depicted by markers on the bottom line. Green circles are attached to a 3 kyr uncertainty (green
horizontal error-bar represented at 117.4 ka BP), purples squares are associated with a 6 kyr uncertainty (purple horizontal



error-bar represented at 354.1 ka BP) and red markers with a 10 kyr uncertainty (red horizontal error-bar represented at 660.7
ka BP). Between 390 and 475 ka BP, all extrema are not tuned to the target due to the poor resemblance between the signal
and insolation.

The uncertainty arising from the filter used and from the tie point identification method can be estimated
by a comparison of the $\delta O_2/N_2$ peak positions identified before and after filtering of the signal with two different
methods (supplementary Fig. S2 and Table S2). The resulting uncertainty is of 700 years on average (with a
standard deviation of 250 years), reaching 2,100 years around 230 ka BP.
The new highly resolved data presented here enable a better description of the signal variability and a
reduction of the uncertainty associated with orbital tie points.

### 3.2.2    Total Air Content

The new TAC record is continuous over the last 800 kyr with a mean sampling resolution of 2,000 years
(Fig. 3). The raw data between 800 and 440 ka BP are not shown here and will be published in a separate study
(Capron et al., in prep). The TAC series shows a good resemblance with the integrated summer insolation (ISI,
obtained by a summation over a year of all daily insolation at 75° South above a chosen threshold). After
comparison in the frequency domain of the TAC record of EDC ice core with ISI obtained using different
thresholds, the ISI curve calculated for a threshold of 375 $Wm^{-2}$ (ISI375) exhibits the finest spectral agreement
with the TAC record of EDC ice core over the past 800 kyr. The coherency between the TAC record and ISI is
deficient over MIS 11 (between 430 and 370 ka BP) and in the deepest part of the core (prior to 700 ka BP) where
the signal to noise ratio is low.
Following a data processing treatment consistent with the method described by Lipenkov et al. (2011),
the 800 kyr long TAC dataset is interpolated every 100 years, and then filtered with a band-pass filter rejecting
periods below 15,000 and above 46,000 years (IgorPro FIR filter with a KaiserBessel20 window: cut-off from 15
to 14 kyr period and from 46 to 47 kyr period, number of coefficients of 559). Outliers in the raw TAC dataset are
discarded if they show an anomaly superior to 1.0 mL/kg (standard deviation of TAC record) when compared to
the band-pass filtered signal. 45 outliers are rejected out of 399 datapoints (among which 16 outliers are identified
between 100 and 0 ka BP). The TAC is interpolated and filtered again after removal of the outliers.
Tie points are mostly determined by matching variations extrema of TAC and integrated summer
insolation at 75°S (see Fig. 3). Indeed, in case of a non-linear relationship between TAC and insolation, extrema
are better indicators of TAC response to insolation forcing. Moreover, filtering the dataset induces a bias in the
mid-slope position. The method employed to determine extrema position is the same as for $\delta O_2/N_2$ insolation tie
points. Only one of the tie points is identified by matching mid-slopes (i.e. derivative extremum) at 362 ka BP
rather than minima at 375 ka BP due to the flatness of the insolation minimum which precludes to identify an
accurate tie point. All extrema are not tuned to the target due to the poor resemblance between the signal and
insolation and 42 unambiguous tie points were kept out of 64 detected by the automated method. The tie point
uncertainty finds its origin in the age errors associated with the filtering (~700 years), tie point identification and
outlier rejection (~900 years). It is evaluated to be 3 kyr when there are good agreements: (i) between the signal
and its target, meaning that one peak in ISI375 is reflected by a singular peak in the TAC record, and (ii) between
the tie points identified by the automated method and manually (age shift < 1,300 years, average value) (see green
circles, Fig. 3). A 6 kyr uncertainty is attached to the tie points if the latter condition is not respected (age shift >



1,300 years) (see purple squares, Fig. 3) and a 10 kyr uncertainty (precession half period) is inferred to the tie
points if the ISI375 variations are not reflected by the TAC record, meaning that one peak in ISI375 could be
associated with two peaks in the TAC record, or if the signal to noise ratio of the TAC record is too large (see red
markers, Fig. 3). The choices of filter and orbital target have no significant impact on the chronological uncertainty,
a further detailed study is thus beyond the scope of this work.

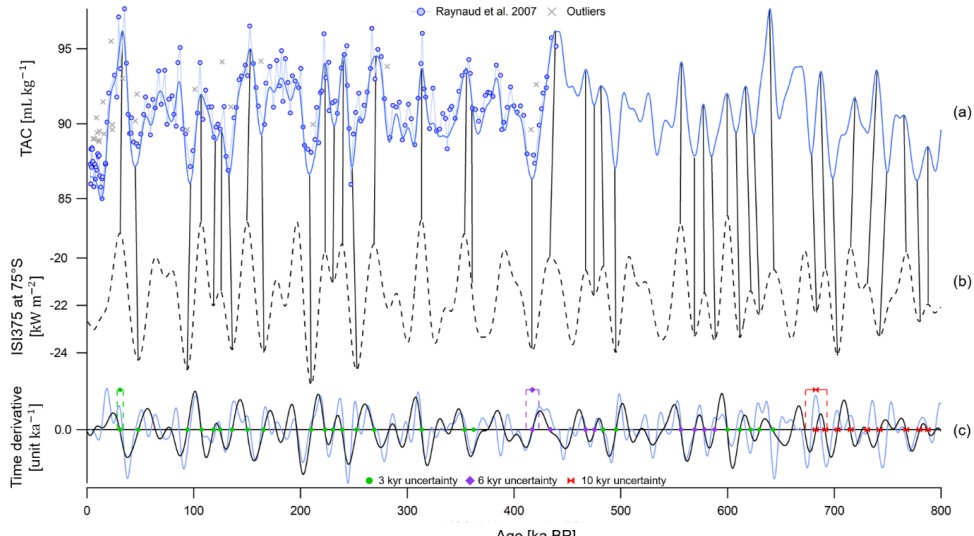

**Figure 3. Alignment of TAC and insolation between 800 and 0 ka BP.** (a) EDC raw TAC data (blue circles, Raynaud et al.
2007), outliers (grey crosses) and filtered signal (blue line) on AICC2012 ice timescale. The raw data between 800 and 440 ka
BP are not shown here and will be published in a separate study (Capron et al., in prep). (b) ISI375 at 75°S on a reversed axis.
The matching peaks and mid-slopes are linked by vertical bars. (c) Time derivative of insolation (black line) and TAC (blue
line). Its 0 value corresponds to extreme values in insolation and TAC. The determined tie points between TAC and insolation
are depicted by markers on the bottom line. Green circles are attached to a 3 kyr uncertainty (green horizontal error-bar
represented at 31 ka BP), purple squares are associated with a 6 kyr uncertainty (purple horizontal error-bar represented at 417
ka BP) and red markers with a 10 kyr uncertainty (red horizontal error-bar represented at 683 ka BP).

The orbital tuning results in 42 new tie points over the last 800 kyr (displayed in Fig. 3 and compiled in Table
S1). They replace the 14 tie points used to constrain EDC ice timescale in AICC2012 between 425 and 0 ka BP,
that were derived by direct matching mid-slope variations of unfiltered TAC and ISI target and attached to an
uncertainty varying between 2.9 and 7.2 kyr.

### 3.2.3    $\delta^{18}O_{atm}$
In this work, new highly resolved $\delta^{18}O_{atm}$ data on EDC ice core are presented over Terminations II, III, IV,
V and VI (Fig. 4). The available $\delta^{18}O_{atm}$ data can be sorted out in two groups: new $\delta^{18}O_{atm}$ data (Grisart, 2023)
at high temporal resolution (between 333 and 1,375 years, see Table 1) and old measurements compiled by Extier
et al. (2018b), characterised by a lower sampling resolution (between 1,000 and 1,500 years, see Table 1), except
between 374 and 153 ka BP (resolution between 160 and 700 years, see Table 1). The new dataset allows us to
improve the resolution of the long EDC record over MIS 5, 7, 9 and in particular over MIS 11 and 13, periods of



sparsity in the ancient record (Extier et al., 2018b). Although the two datasets agree globally well over the last 800
kyr, the new highly resolved dataset refines the signal between 255.5 and 243 ka BP where a lot of noise is present
in the record of Extier et al. (2018b) (see inset in Fig. 4). This noise may be explained by the fact that highly
resolved (mean sampling resolution of 381 years) measurements were performed on ice samples stored at -20℃
in the compilation by Extier et al. (2018b) while the new measurements are performed on ice stored at -50℃.
Therefore, we chose to remove the noisy dataset of Extier et al. (2018b) between 255.5 and 243 ka BP before
combining the novel dataset with the remaining 800 kyr long record of Extier et al. (2018b).

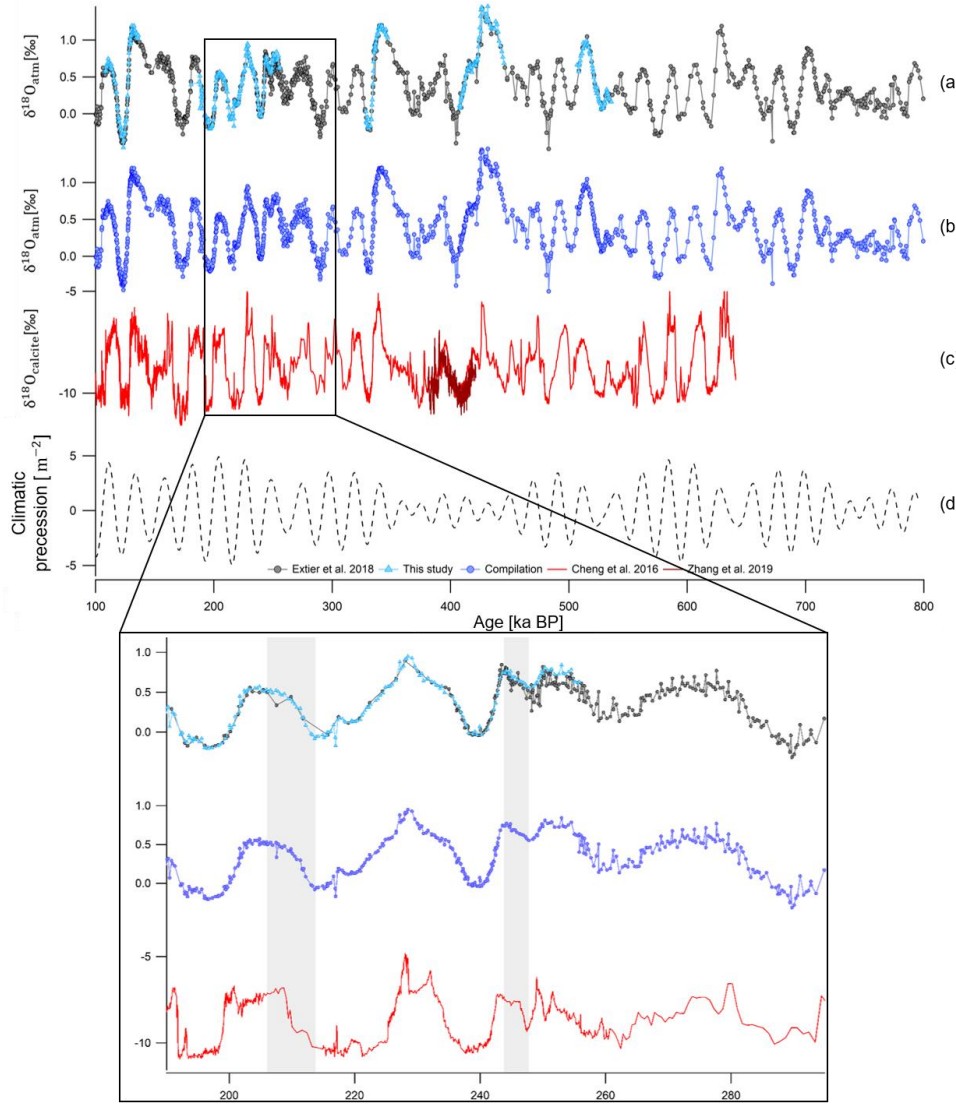


**Figure 4. Evolution of EDC $\delta^{18}O_{atm}$ record between 800 and 100 ka BP.** (a) EDC $\delta^{18}O_{atm}$ raw old data (black circles,
Extier et al., 2018b) and EDC $\delta^{18}O_{atm}$ raw new data (blue triangles, Grisart, 2023) on AICC2012 gas timescale. (b)
Compilation of the two datasets after removal of old measurements between 255.5 and 243 ka BP. (c) $\delta^{18}O_{calcite}$ composite




record from speleothems from Sambao, Dongge, Hulu (red line) and Yongxing (brown line) caves (Zhao et al., 2019; Cheng et al., 2016). (d) Climatic precession from Laskar et al. (2004) delayed by 5,000 years. Inset is a zoom between 295 and 190 ka BP. Grey vertical bars highlight the improved agreement between new data of Grisart (2023) (blue triangles) and $\delta^{18}O_{calcite}$ (red line) than between old data (grey circles) and $\delta^{18}O_{calcite}$.

Following the dating approach proposed by Extier et al. (2018), $\delta^{18}O_{atm}$ and $\delta^{18}O_{calcite}$ are aligned using mid-slopes of their variations over the last 640 kyr. To do so, the compiled EDC $\delta^{18}O_{atm}$ record and the Chinese $\delta^{18}O_{calcite}$ signal are linearly interpolated every 100 years, smoothed (25 points Savitzky-Golay) and extrema in their temporal derivative are aligned. It should be specified that synchronising $\delta^{18}O_{atm}$ and East Asian $\delta^{18}O_{calcite}$ is not always obvious due to the long residence time of oxygen in the atmosphere (1-2 kyr) which may not be compatible with $\delta^{18}O_{calcite}$ abrupt variations over glacial inceptions and terminations. In particular, the slow increase of the $\delta^{18}O_{atm}$ record from 370 to 340 ka BP does not resemble the evolution of $\delta^{18}O_{calcite}$ which is first moderate then abrupt over the same period (Fig. 5, red area). For this reason, we chose not to use the two tie points identified by Extier et al. (2018) at 351 and 370.6 ka BP. The new highly resolved data enable to identify five new tie points and to shift five tie points that have been determined beforehand by Extier et al. (2018) (Fig. 5). Between 248 and 244 ka BP, the new $\delta^{18}O_{atm}$ measurements do not coincide with the $\delta^{18}O_{calcite}$ variations and we decided to remove the tie point identified by Extier et al. (2018) at 245.4 ka BP (Figure 5, red area). Between 480 and 447 ka BP, the $\delta^{18}O_{atm}$ variations are characterized by a low resolution (1.1 kyr) and a weak amplitude, which prevents unambiguous matching of $\delta^{18}O_{atm}$ and $\delta^{18}O_{calcite}$. The four tie points identified by Extier et al. (2018) at 447.3, 449.9, 455.9 and 462.8 ka BP are thus rejected (Fig. 5, red area). The remaining 39 tie points defined by Extier et al. (2018) are preserved and used here to constrain EDC gas age. Their uncertainty varies between 1.1 and 7.4 kyr.



**Figure 5. Alignment of EDC $\delta^{18}O_{atm}$ and Chinese $\delta^{18}O_{calcite}$ records over time periods where new tie points are defined.** (a) EDC $\delta^{18}O_{atm}$ new and old datasets. (b) Compiled EDC $\delta^{18}O_{atm}$. (c) Chinese $\delta^{18}O_{calcite}$. (d) Temporal derivatives of compiled EDC $\delta^{18}O_{atm}$ (blue curve) and of the old $\delta^{18}O_{atm}$ dataset (black curve). (e) Temporal derivative of Chinese $\delta^{18}O_{calcite}$ (red curve). Extrema in temporal derivatives are aligned. New tie points are represented by blue vertical bars and tie points determined by Extier et al. (2018) and used in the new AICC2023 chronology by black vertical bars. Dotted vertical



bars show tie points identified by Extier et al. (2018) that are not used in AICC2023. Red vertical areas frame periods of lacking
resemblance between $\delta^{18}O_{atm}$ and $\delta^{18}O_{calcite}$ variations.

Between 810 and 590 ka BP, we updated the following approach of Bazin et al. (2013): EDC $\delta^{18}O_{atm}$
and 5 kyr delayed climatic precession are synchronized using mid-slopes of their variations. However, $\delta^{18}O_{atm}$
should rather be aligned to precession without delay when no Heinrich-like events occurs. Indeed, $\delta^{18}O_{atm}$ is
sensitive to both orbital and millennial scale variations of the low latitude water cycle (Capron et al., 2012a;
Landais et al., 2010) and Heinrich-like events occurring during deglaciations delay the response of $\delta^{18}O_{atm}$ to
orbital forcing through southward ITCZ shifts (Extier et al., 2018). We thus chose to align $\delta^{18}O_{atm}$ to precession
when no Ice Rafted Debris (IRD) peak is visible on the studied period and keep a 5 kyr delay when IRD peaks are
identified. This results in shifting 12 tie points of Bazin et al. (2013) by 5,000 years towards older ages (see Fig.
6). The eight remaining tie points of Bazin et al. (2013). that coincide with peaks in the IRD record of Barker
(2021) are kept (Fig. 6). A 6 kyr uncertainty is attributed to the $\delta^{18}O_{atm}$ derived tie points over the period between
810 and 590 ka BP.
69 new $\delta^{18}O_{atm}$ tie points are determined here over the last 810 kyr (displayed in Fig. 5 and 6 and
compiled in Table S1). They are attached to an uncertainty varying between 1.1 and 7.4 kyr and replace the 39 tie
points used to constrain EDC gas timescale in AICC2012 between 800 and 363 ka BP (Bazin et al., 2013). The
same alignment method is applied between Vostok $\delta^{18}O_{atm}$ (Petit et al., 1999) and Chinese $\delta^{18}O_{calcite}$ and 36 new
tie points are determined (see Supplementary Material), replacing the 35 tie points used to constrain Vostok gas
timescale in AICC2012.
Finally, there was a redundancy in the dating of the bottom part of the EDC ice core in AICC2012 where
both $\delta^{18}O_{atm}$ orbital tie points and [10]Be peaks corresponding to the Matuyama - Brunhes geomagnetic reversal
event were used. Indeed, the two [10]Be dating constraints at 780.3 and 798.3 ka BP were directly derived from the
$\delta^{18}O_{atm}$ orbital dating and not obtained independently (Dreyfus et al., 2008). We thus decided to remove the [10]Be
age constraints.



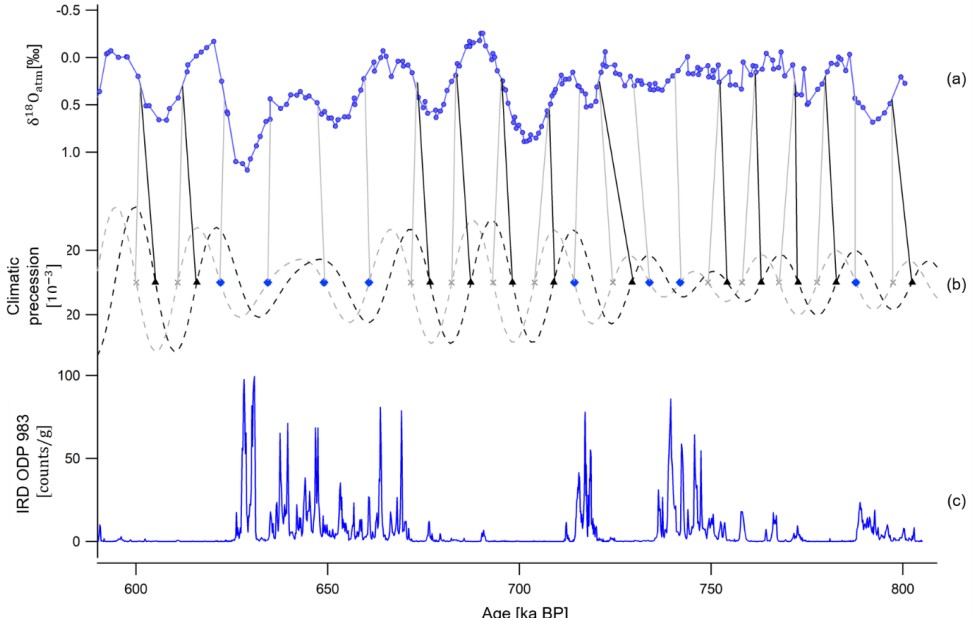

**Figure 6. Alignment of EDC $\delta^{18}O_{atm}$ and climatic precession between 810 and 590 ka BP.** (a) Compiled EDC $\delta^{18}O_{atm}$ on AICC2012 gas timescale. (b) Precession delayed by 5,000 years (grey dashed line) and not delayed (black dashed line) (Laskar et al., 2004). (c) Ice-Rafted Debris at ODP983 site (North Atlantic Ocean, southwest of Iceland) by Barker (2021). Grey crosses represent mid-slopes of 5 kyr delayed precession variations. Grey vertical bars illustrate tie points between precession and EDC $\delta^{18}O_{atm}$ used by Bazin et al. (2013) to constrain AICC2012 gas timescale. Black triangles show mid-slopes of variations of precession without delay. Black verticals bars display new tie points between precession and EDC $\delta^{18}O_{atm}$ established when no Heinrich-like events is shown by IRD record. Blue squares represent mid-slopes of 5 kyr delayed precession variations when Heinrich-like events are shown by IRD record. Black and blue markers correspond to the new tie points defined in this study.

## 3.3 Background scenario of LID

In this work, new highly resolved data $\delta^{15}N$ on EDC ice core are presented over Terminations II, III, IV, V and VI (Fig. 7a). The available $\delta^{15}N$ data can be sorted out in two groups: $\delta^{15}N$ measured by Grisart (2023) and Bréant et al. (2019) at high temporal resolution (between 333 and 1,375 years, see Table 1) and the older measurements (Bazin et al., 2013) used to estimate LID in AICC2012, characterised by a lower sampling resolution (between 1,400 and 2,400 years, see Table 1). The measurements of Bazin et al. (2013) and Bréant et al. (2019) have been shifted down by 0.04 ‰ to account for calibration errors. The new dataset permits to extend the record around 1100 m and between 1700 and 2500 m and to improve the resolution over Terminations II to VI.



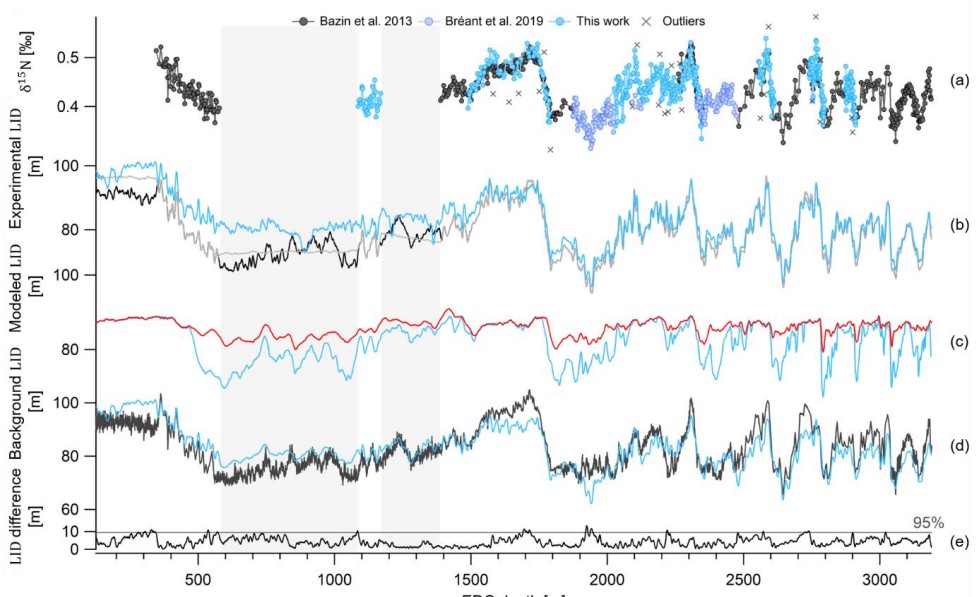

**Figure 7. EDC $\delta^{15}$N record and past LID evolution as a function of EDC depth**. (a) New and highly-resolved $\delta^{15}$N dataset (blue circles), old dataset (black circles) and outliers (rejection criterion of 1σ) (grey crosses). (b) LID calculated as per LID ≃ $\delta^{15}N_{grav} \cdot \frac{RT}{g}$ for 3 cases: 1) $\delta^{15}N_{grav} = \delta^{15}$N with the $\delta^{15}$N record constructed by interpolation between data when no data are available (grey), 2) $\delta^{15}N_{grav} = \delta^{15}$N with the $\delta^{15}$N record constructed by normalisation of the δD record when no data are available (black ), 3) $\delta^{15}N_{grav} = \delta^{15}$N − $\delta^{15}N_{therm}$ with $\delta^{15}N_{therm}$ estimated by the firn model (Bréant et al., 2017) and the $\delta^{15}$N record constructed by interpolation between data when no data are available (blue). (c) Modeled LID according to test 1 (with impurity concentration, blue) and test 2 (without impurity concentration, red) (see Table 3). (d) Background scenarios of LID used to construct AICC2012 (black) and inputs in Paleochrono to obtain AICC2023 (blue). (e) Absolute difference between prior LID of AICC2012 and AICC2023. The grey line separates the 5% highest values from the rest. The grey rectangles cover areas when no $\delta^{15}$N data are available.

Outliers are discarded if they show an anomaly superior to 0.045 ‰ when compared to the smoothed record (Savitzky-Golay algorithm with 25 points). This results in the rejection of 25 datapoints out of 475 measurements for the new dataset (see Fig. 7). The two $\delta^{15}$N datasets are merged and the compiled record is interpolated every 100 years. Then, assuming that the firn is solely a diffusive zone (i.e. no convection layer at the top) at EDC during the last 800 kyr, in agreement with current observations (Landais et al., 2006), past LID is calculated as per the first order estimate of the barometric equation:

$$LID \simeq \delta^{15}N_{grav} \cdot \frac{RT}{g} \qquad (5)$$

with $T$ the temperature at EDC estimated from combined measurements of ice $\delta^{18}$O and δD after correction of the influence of the sea water $\delta^{18}$O (Landais et al., 2021).

In absence of a large thermal gradient within the firn (mostly present in Greenlandic ice cores during Dansgaard Oeschger events), $\delta^{15}$N is mainly modulated by gravitational fractionation of $N_2$ molecules occurring from the surface down to the lock-in zone, and $\delta^{15}$N measured in bubbles hence approximately reflects the LID



(Landais et al., 2006; Severinghaus et al., 1996) and $\delta^{15}N_{grav} \simeq \delta^{15}N$ in Eq. (5) (grey and black lines in Fig. 7b).
To account for a small temperature gradient in the firn in Antarctic ice core, the thermal fractionation term
$\delta^{15}N_{therm}$ can be estimated by the firn model (Bréant et al., 2017). Past LID is then calculated as per Eq. (5) with
$\delta^{15}N_{grav} = \delta^{15}N - \delta^{15}N_{therm}$ (blue curve in Fig. 7b). Thermal fractionation represents a maximum correction of
4.2 m on the LID at EDC.
When $\delta^{15}N$ measurements are not available, Bazin et al. (2013) used a synthetic $\delta^{15}N$ signal based on the
correlation between $\delta^{15}N$ and $\delta D$ to estimate the LID background scenario (black curve in Fig. 7b). Indeed, for
different Antarctic sites, it has been observed that $\delta^{15}N$ and $\delta D$ are well correlated over the last Termination on a
coherent timescale (Dreyfus et al., 2010; Capron et al., 2013). Since then, Bréant et al. (2019) presented new high
resolution measurements of $\delta^{15}N$ extending the signal over Termination III (around 2300 m, 250 ka BP). Their
study unveiled the anatomy of this atypical deglaciation: the interplay between Heinrich-like events and bipolar
seesaw mechanism induced a strong warming of Antarctic temperature, resulting in divergent $\delta^{15}N$ and $\delta D$
records. Therefore, using $\delta D$ to construct a synthetic $\delta^{15}N$ scenario should be done carefully. For this reason, the
firn densification model described in Bréant et al. (2017) is employed to estimate LID evolution in the past when
$\delta^{15}N$ data are missing, rather than using the $\delta D - \delta^{15}N$ relationship, as it was done for AICC2012. After different
sensitivity tests, we choose to keep the parameterization preferred by Bréant et al. (2017): test 1 in Table 3 (i.e.
firn densification activation energy depending on the temperature and the impurity concentration) as it is believed
to give the most probable evolution of LID over the last 800 kyr (see Supplementary Material).
The final LID background scenario is calculated as a function of EDC depth (Table 5, Fig. 7d). To obtain
a coherent scenario, the firn modeling estimates have been adjusted, by standard normalization, to the scale of LID
values derived from $\delta^{15}N$ data (later referred to as experimental LID). The final LID scenario has been smoothed
using a Savitzky-Golay algorithm (25 points), and then provided as an input file to Paleochrono.

**Table 5. Method of determination of LID background scenario according to EDC depth range.** The thermal fractionation
term is estimated by the firn model running in the same configuration as for calculating the modeled LID, i.e. Test 1 (Table 3):
Firn densification activation energy depending on the temperature and impurity concentration.

| Depth range (m) | 0 – 345 | 345 - 578 | 578 - 1086 | 1086 – 1169 | 1169 – 1386 | 1386 – Bottom |
|---|---|---|---|---|---|---|
| $\delta^{15}N$ data availability | No | Yes | No | Yes | No | Yes |
| Method of determination of the LID | From constant $\delta^{15}N$ (measured at 345 m) and corrected for thermal fractionation. | From $\delta^{15}N$ data, corrected for thermal fractionation and smoothed. | From firn modeling and scaled to experimental LID values. | From $\delta^{15}N$ data, corrected for thermal fractionation and smoothed. | From firn modeling and scaled to experimental LID values. | From $\delta^{15}N$ data, corrected for thermal fractionation and smoothed. |




The other necessary input files for Paleochrono, Accumulation ($A$) and Thinning ($\tau$) background scenarios,
are the same as in Bazin et al. (2013). $A$ is estimated from water isotopes (Parrenin et al., 2007b) and $\tau$ from
unidimensional ice-flow modeling (Parrenin et al., 2007a).

### 3.4 New stratigraphic links between EDC and other ice cores

EDC can be linked to other ice cores via ice and gas stratigraphic links identified during abrupt climate
changes recorded in Greenlandic and Antarctic ice cores. To establish AICC2012, Bazin et al. (2013) used 255
gas stratigraphic tie points coming from the matching of $CH_4$ (or $\delta^{15}N$ when $CH_4$ is not available at NGRIP) or
$\delta^{18}O_{atm}$ variations between EDC, EDML, Vostok, NGRIP and TALDICE. Here we revise the majority of these
tie points using the synchronization of $CH_4$ series of EDC, Vostok and TALDICE to up-to-date highly resolved
records from EDML and NGRIP ice cores over the last interglacial offset and the last glacial period (Baumgartner
et al., 2014). From 122 to 10 ka BP, Baumgartner et al. (2014) identified 39 stratigraphic links between EDML
and NGRIP by matching mid-points of the $CH_4$ abrupt changes with a precision of 300 to 700 years. When they
also detected such rapid variations in lower resolved $CH_4$ records of TALDICE, Vostok and EDC ice cores, they
extended the stratigraphic links to the five ice cores but assigned them a larger uncertainty (up to 1,500 years).
AICC2012 was further constrained by 534 ice stratigraphic links identified from volcanic matching and
synchronization of cosmogenic isotopes between the five ice cores. Here we replace some of the stratigraphic links
between NGRIP, EDML and EDC by highly resolved volcanic matching points (Svensson et al., 2020). The
application of volcanic proxies and annual layer counting helped them identify large volcanic eruptions that left a
specific signature in both Greenland and Antarctica. Such signature is defined by sulfate patterns (indicating
singular volcanic events separated by the same time interval in ice cores from both poles). Their study spotted 82
large bipolar volcanic eruptions over the second half of the last glacial period (from 60 to 12 ka BP), providing as
many ice stratigraphic links synchronising EDC with EDML and EDML with NGRIP within a small relative
uncertainty (i.e. ranging from 1 to 50 years, of 12 years on average). Between 43 and 40 ka BP, five cosmogenic
tie points associated with the Laschamp geomagnetic excursion (Raisbeck et al., 2017) replace the volcanic
matching over this period (Svensson et al., 2013), shifting the tie points by ∼30 years.

## 4    Discussion

### 4.1 New AICC2023 chronology

#### 4.1.1    Impact of absolute age constraints

A large uncertainty is linked with $^{81}Kr$ dating, therefore $^{81}Kr$ age estimates do not significantly change
the chronology (maximum 200 years) (see Fig. 8). $^{81}Kr$ age estimates are systematically older than the new
timescale (by 25 to 36 kyr, see Fig. 9). This observation could also indicate an undervaluation of $^{81}Kr$ half-life.

#### 4.1.2    Consistency between orbital age constraints

To evaluate the consistency between the orbital age constraints, several "test chronologies" were
produced. Each "test chronology" of EDC ice core was obtained by running one multi-site (EDC, Vostok, EDML,
TALDICE, NGRIP) experiment of Paleochrono. In each of these tests, we implement one category of new age
constraints presented in this work while keeping AICC2012 parameters for other categories. Several "test
chronologies" are thus constructed: the $^{81}Kr$, $\delta O_2/N_2$, TAC, $\delta^{18}O_{atm}$ and stratigraphic links based chronologies



(Fig. 8). Two additional "test chronologies" were obtained by implementing and modifying age constraints either
on Vostok or TALDICE to the AICC2012 chronology as explained in Sect. 2.a (Fig. 8, dotted lines). EDC ice age
difference between each "test chronology" and the AICC2012 timescale is represented in Fig. 8 so that it is possible
to read which type of dating tool suggests to shift the background chronology towards either older or younger ages.

Although the three orbital dating tools globally agree with each other over the last 800 kyr, meaning that

they all tend to shift the background chronology towards either older or younger ages over a certain period of time,
they sometimes are inconsistent (see Fig. 8). The three largest inconsistencies involve age differences between
$\delta O_2/N_2$, TAC and $\delta^{18}O_{atm}$ based chronologies reaching 4.15 to 8.3 kyr (Table 6). At 390 ka BP, an 8.3 kyr large
discrepancy is observed between $\delta O_2/N_2$ and $\delta^{18}O_{atm}$ based chronologies. Over this period, the low resolution
$\delta O_2/N_2$ record variations do not match its orbital target variations (two insolation minima against one $\delta O_2/N_2$
maximum, see Fig. 1). For this reason, the $\delta O_2/N_2$ age constraints identified between 480 and 350 ka BP were
attached to a 6 kyr uncertainty (quarter of a recession period, Fig. 2). In contrast, the $\delta^{18}O_{atm}$ record agrees well
with $\delta^{18}O_{calcite}$ (Fig. 4) and the uncertainty attached to the $\delta^{18}O_{atm}$ inferred tie points over this interval is smaller.
Hence, the new AICC2023 chronology suggests to shift AICC2012 towards older ages by 2.2 kyr, as per the
$\delta^{18}O_{atm}$ based chronology (Fig. 8). Around 550 ka BP, the TAC and $\delta^{18}O_{atm}$ based chronologies strongly diverge.
This may be caused by the absence of TAC tie points due to the non-coincidence of TAC and ISI375 extrema (Fig.
3) while there is a good agreement between $\delta^{18}O_{atm}$ and $\delta^{18}O_{calcite}$ records. Therefore, we decide to increase up
to 6 kyr the uncertainty attached to the four TAC age constraints between 600 and 550 ka BP (Fig. 3) and
AICC2023 is rather following the $\delta^{18}O_{atm}$ based chronology, inducing older ages than AICC2012. At 765 ka BP,
the discordance between $\delta O_2/N_2$ (and TAC) and $\delta^{18}O_{atm}$ based chronologies is likely due to the poor quality of
the records from the lowermost part of the core. Over these oldest time periods, $\delta^{18}O_{atm}$, TAC and $\delta O_2/N_2$ were
tied up respectively with precession, integrated insolation and insolation with a large uncertainty (6 to 10 kyr).
This leads to a final chronology AICC2023 suggesting a larger chronological uncertainty than AICC2012 as well
as younger ages (as per TAC and $\delta O_2/N_2$ chronologies) over MIS 18, and then older ages (as per $\delta^{18}O_{atm}$
chronology) over MIS 19.











**Table 6. Description of the inconsistencies between TAC, $\delta O_2/N_2$ and $\delta^{18}O_{atm}$ based chronologies.** The age shift suggested by each dating tool with respect to AICC2012 age is detailed. The age position of the disagreement is given as per AICC2023. We did not highlight inconsistencies between TAC and $\delta O_2/N_2$ based chronologies as they remain within their respective orbital uncertainty.

| | $\delta^{18}O_{atm}$ | | |
|---|---|---|---|
| $\delta O_2/N_2$ | Non-coherent | Non-coherent | Non-coherent |
| TAC | Coherent | Non-coherent | Non-coherent |
| **Disagreement type** | $\delta O_2/N_2$ chronology younger by 4,700 years than AICC2012 | TAC (and $\delta O_2/N_2$) chronology younger by 2,400 (and 800) years than AICC2012 | $\delta O_2/N_2$ (and TAC) chronology younger by 2,850 (and 1,300) years than AICC2012 |
| | $\delta^{18}O_{atm}$ chronology older by 3,600 years than AICC2012 | $\delta^{18}O_{atm}$ chronology older by 1,700 years than AICC2012 | $\delta^{18}O_{atm}$ chronology older by 2,800 years than AICC2012 |
| **Interval of disagreement (ka BP)** | 390 (MIS 11) | 550 (MIS 14) | 765 (MIS 19) |

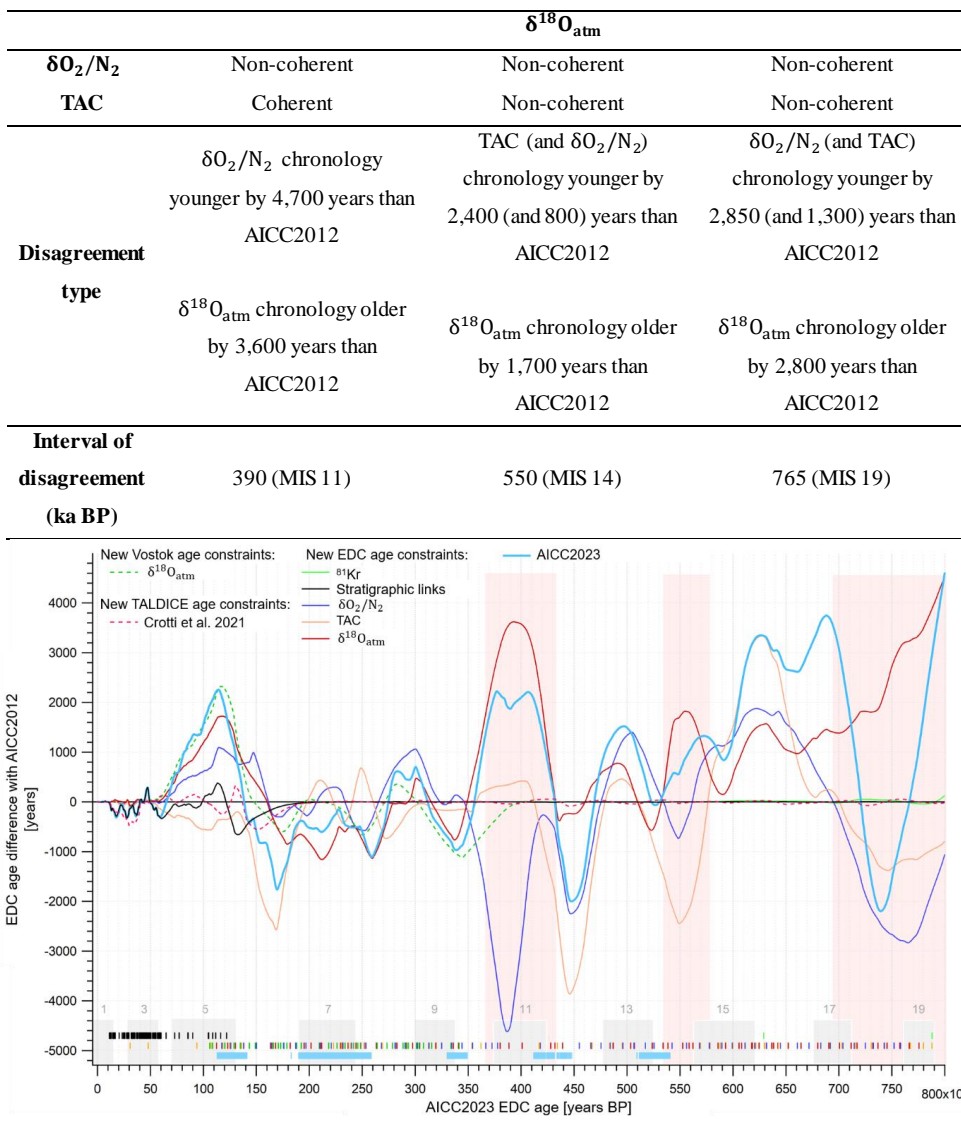

**Figure 8. EDC ice age difference between AICC2012 and different tests chronologies obtained with Paleochrono over the last 800 kyr.** The ice age difference is calculated as per ("test chronology" – AICC2012). Two "test chronologies" are obtained either by addition of new Vostok $\delta^{18}O_{atm}$ – $\delta^{18}O_{calcite}$ age constraints (green dotted line) or of stratigraphic and absolute TALDICE constraints between 470 and 129 ka BP from Crotti et al. (2021) (red dotted line). The other "test chronologies" are constructed by implementing either: 1) $^{81}$Kr (green), 2) $\delta O_2/N_2$ (dark blue), 3) TAC (orange), 4) $\delta^{18}O_{atm}$ (red) and 5) stratigraphic links with NGRIP, EDML, TALDICE, Vostok (black) to replace AICC2012 constraints. Vertical bars represent the corresponding age horizons. AICC2023 is obtained by implementing the new constraints all together (light blue



line). Light blue vertical bars show new data collected by Grisart (2023) and presented in this work. The three largest
inconsistencies between $\delta O_2/N_2$, TAC and $\delta^{18}O_{atm}$ chronologies are shown by red areas. Grey squares indicate interglacials
from MIS 19 to MIS 1.
### 4.1.3    Final chronology and uncertainty

The new AICC2023 chronology suggests significant age shifts when compared to AICC2012 over old
periods, including 3.8 and 5 kyr shifts towards older ages around 800 and 690 ka BP as well as a 2.1 kyr shift
towards younger ages around 730 ka BP. The chronology is also strongly modified over MIS 5 and MIS 11 where
AICC2023 is about 2 kyr older than AICC2012. These 2 kyr shifts are induced by $\delta O_2/N_2$, $\delta^{18}O_{atm}$ dating
constraints and stratigraphic links over MIS 5 and by $\delta^{18}O_{atm}$ and TAC constraints over MIS 11. When averaged
over the past 800 kyr, the chronological uncertainty is reduced from 2.5 kyr for AICC2012 to 1.8 kyr here. Still, it
remains significant (above 2 kyr) over MIS 11 and in the lowermost part of the core, between 800 and 650 ka BP.
Specifically, between 800 and 670 ka BP, the uncertainty associated with the new AICC2023 timescale sometimes
is larger than the AICC2012 uncertainty (Fig. 9). This is caused by a larger relative error attached to TAC and
$\delta O_2/N_2$ age constraints as well as by the eviction of the two redundant [10]Be age constraints at 780.3 and 798.3 ka
BP associated with the Matuyama - Brunhes geomagnetic reversal event.

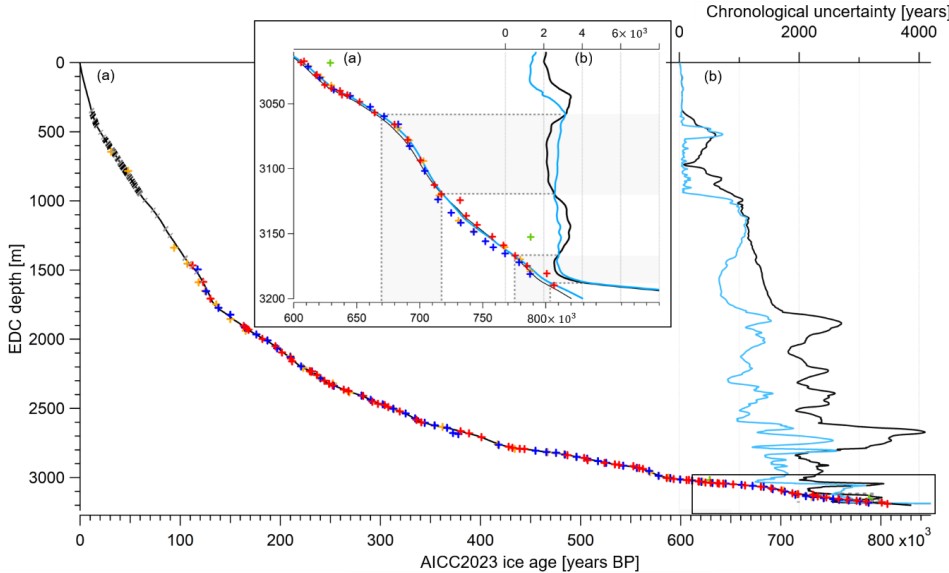

**Figure 9. EDC ice age and uncertainty as a function of the depth.** (a) EDC ice age (AICC2012 in black, AICC2023 in blue).
(b) Uncertainty (AICC2012 in black, AICC2023 in blue). Crosses and slashes represent new age constraints (ice stratigraphic
links in black, gas stratigraphic links in grey, $\delta^{18}O_{atm}$ in red, $\delta O_2/N_2$ in blue, TAC in orange, [81]Kr in green). Inset is a zoom
in between 800 and 600 ka BP. Grey rectangles frame periods where the new AICC2023 uncertainty is larger than AICC2012
uncertainty.

The age difference between ice and gas timescales ($\Delta$age) is of 3 kyr on average, reaching its largest value
(5 kyr) during the cold eras of MIS 12, 8, 6 and 4 (at 440, 260, 145 and 70 ka BP respectively, Fig. 10). A 4 kyr
$\Delta$age is obtained at around 160 ka BP (Fig. 10), consistent with the use of new $\delta^{15}N$ data of Bréant et al. (2019)



leading to a background scenario of LID that is 13 m smaller than the prior LID scenario used in AICC2012
between the depths of 1900 and 2000 m (Fig. 7). Using the definition of an interglacial period implying an EDC
δD value surpassing the threshold of - 403 ‰ (EPICA Community members, 2004), we identify ten substages of
interglacials (MIS 1, 5e, 7a, 7c, 7e, 9e, 11c, 15a, 15e and 19, Fig. 10). The average duration of these substages is
reduced by 320 years with the new AICC2023 timescale in comparison with the AICC2012 chronology (Fig. 10).
More specifically, MIS 5e to 15a are shorter while only MIS 15e and MIS 19 are longer. The largest decreases in
duration affecting the Last Interglacial (MIS 5e) and MIS 11.c whose lengths are decreased from 16.3 to 15.1 kyr
and from 31.1 to 30.1 kyr respectively, in agreement with the durations of 14.8 and 29.7 kyr proposed by Extier et
al. (2018).

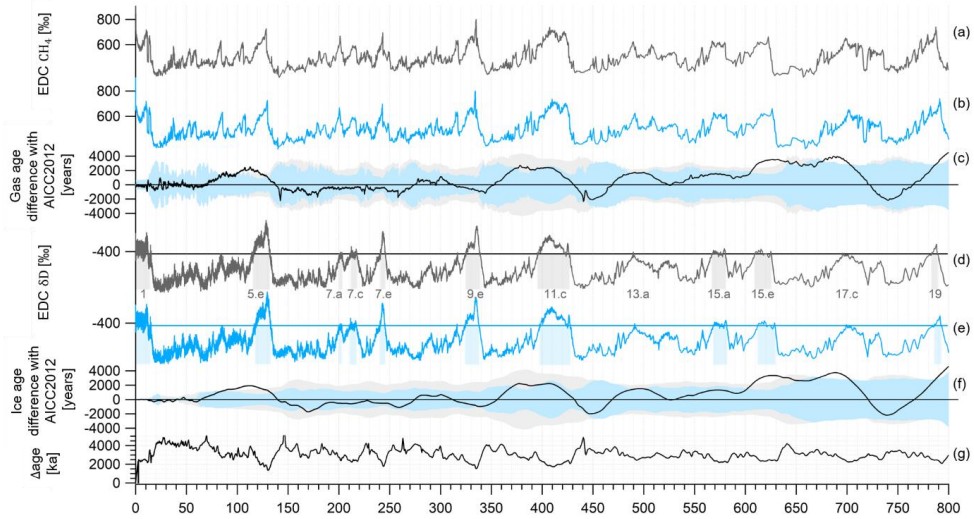

**Figure 10. EDC gas and ice records on AICC2023 (blue) and AICC2012 (black) timescales over the last 800 kyr.** (a)
EDC $CH_4$ on AICC2012 and (b) AICC2023 gas timescales. (c) Gas age difference AICC2023 – AICC2012. Grey and blue
envelops are AICC2012 and AICC2023 chronological uncertainties respectively. (d) EDC δD on AICC2012 and (e) AICC2023
ice timescales. Grey and blue rectangles indicate interglacial periods defined when δD is superior to the threshold of - 403 ‰
(horizontal lines) (EPICA members, 2004). Interglacials are numbered from MIS 1 to 19 (Berger et al., 2016). (f) Ice age
difference AICC2023 – AICC2012. (g) Age difference between ice and gas AICC2023 timescales (Δage).

**4.2 Comparison with other chronologies**
**4.2.1    MIS 5 (from 130 to 80 ka BP)**

When Veres et al. (2013) presented the AICC2012 chronology over the last climatic cycle, they identified

a disagreement with the Greenland timescale GICC05-modelext between 115 and 100 ka BP. The comparison
between the Greenland $\delta^{18}O_{ice}$ record and the $\delta^{18}O_{calcite}$ from U-Th dated Alpine speleothems shown a delay up
to 2.7 kyr during the Dansgaard Oeschger (D-O) events 23, 24 and 25. Later, this disagreement between abrupt
changes in $\delta^{18}O_{ice}$ from NGRIP (Greenland surface temperature) and $\delta^{18}O_{calcite}$ from the Alps has been re-



evaluated based on a new ice core chronology. Extier et al. (2018) presented a better agreement between the two
records with an older NGRIP timescale than AICC2012 by ~2,200 years for D-O 23 to 25.
On Fig. 11, NGRIP $\delta^{18}O_{ice}$ record is represented on the AICC2023 timescale and is compared to ancient and novel
records of $\delta^{18}O_{calcite}$ from Alpine speleothems (Boch et al., 2011; Moseley et al., 2020). Thanks to new $\delta O_2/N_2$
and $\delta^{18}O_{atm}$ age constraints, the new AICC2023 chronology is also older than AICC2012 between 115 and 100
ka BP and leads to an improved agreement between the records along with a reduction of the uncertainty. This
amelioration is particularly visible over D-O warmings 23 and 24 where the difference between NALPS and
NGRIP chronologies is reduced from ~2,000 years (AICC2012) to 430 and 325 years (AICC2023) respectively
(Table 7).
The Greenland Interstadial (GI) 25 can be subdivided in three substages: GI-25a-b-c with GI-25a the
earliest glacial so-called "rebound event" (Capron et al., 2010). This latter consists in a brief warm-wet excursion
during the slow cooling trend of the longer GI-25 period, before jumping back to a cool-dry climate. The GI-25a
warm-wet interval corresponds to a temperature increase in Greenland and continental Europe and hence identified
by a positive excursion in NGRIP and NALPS $\delta^{18}O$ records (D-O 25 rebound) (Boch et al., 2011; Capron et al.,
2012). At lower latitudes, this rebound likely affected the rainfall amount variations, as exhibited by the abrupt
decrease in the $\delta^{18}O_{calcite}$ from a U-Th dated Sardinian stalagmite from Bue Marino Cave (BMS1, Columbu et
al., 2017). The 2 kyr shift of the new AICC2023 chronology towards older ages permits to improve the coherency
between NALPS, NGRIP and BMS1 timescales over the GI-25a onset (traceable in the $\delta^{18}O$ series, Fig. 11). The
age discrepancy is reduced from ~3,600 years (between AICC2012 and BMS1 timescale) to 1,640 years (between
AICC2023 and BMS1 timescale, Table 7).
**Table 7. Timing of D-O warmings 23 and 24 and D-O 25 rebound event onset.** The GICC05-modelext age uncertainty is
undetermined.

| Event | Timing (a BP) and error (years) | | | | | |
|---|---|---|---|---|---|---|
| | NGRIP ice core timescale | | | | Speleothem timescale | |
| | GICC05-modelext (Wolff et al., 2010) | AICC2012 (Veres et al. 2013) | Extier et al. (2018) | AICC2023 (This study) | BMS1 (Columbu et al. 2017) | NALPS (Boch et al. 2021) |
| D-O 23 warming | 103 995 | 101 850 ± 1310 | 104 090 ± 1200 | 103 980 ± 930 | Not recorded | 103 550 ± 375 |
| D-O 24 warming | 108 250 | 105 850 ± 1330 | 108 010 ± 1200 | 107 975 ± 850 | Not recorded | 108 300 ± 450 |
| D-O 25 rebound onset | 110 960 | 108 100 ± 1410 | 110 280 ± 1200 | 110 120 ± 900 | 111 760 ± 450 | 111 780 ± 630 |


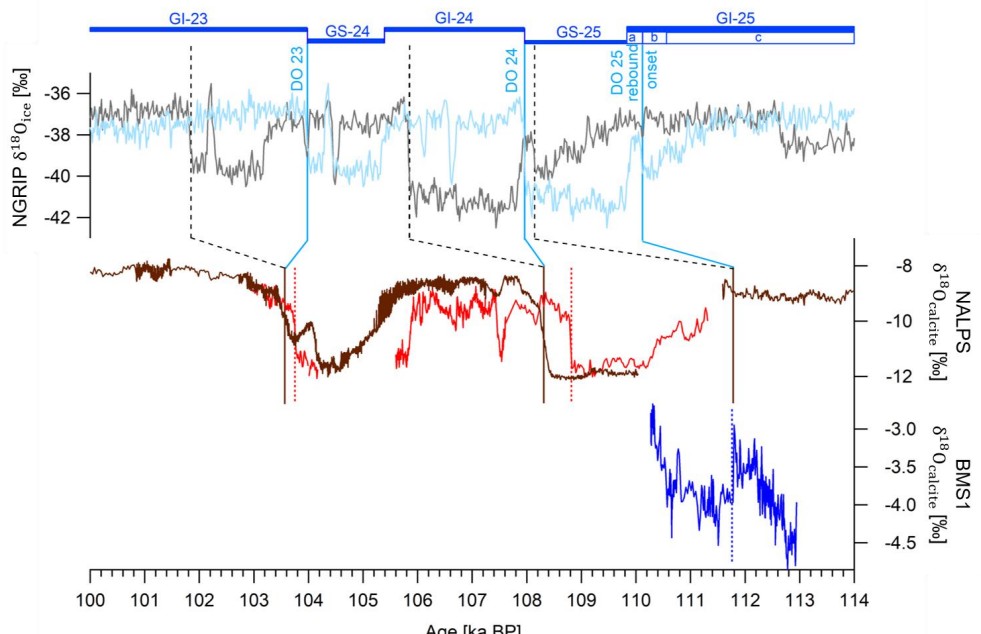

**Figure 11. Northern Alpine speleothems (NALPS) and Bue Marino Stalagmite (BMS1) $\delta^{18}O_{calcite}$ records and NGRIP $\delta^{18}O_{ice}$ evolution between 114 and 100 ka BP.** NGRIP $\delta^{18}O_{ice}$ data by Andersen et al. (2004) on AICC2012 (grey) and AICC2023 (blue) chronologies. NALPS $\delta^{18}O_{calcite}$ data by Moseley et al. (2020) (red) and Boch et al. (2011) (brown). BMS1 $\delta^{18}O_{calcite}$ data by Colombu et al. (2017) (dark blue). Vertical bars indicate D-O 23, D-O 24 and D-O 25 rebound warmings at the onset of GI-23, GI-24 and GI-25a warm-wet substage. They correspond to abrupt increases in the NALPS $\delta^{18}O_{calcite}$ and NGRIP $\delta^{18}O_{ice}$ records and to a decrease in the BMS1 $\delta^{18}O_{calcite}$ series (for the GI-25a onset). Black dashed bars and blue bars show increases in $\delta^{18}O_{ice}$, respectively on AICC2012 and AICC2023 chronologies. Brown bars and red dotted bars show increases in NALPS $\delta^{18}O_{calcite}$ datasets. The blue dotted bar indicates the decrease in BMS1 $\delta^{18}O_{calcite}$. GI/GS (Greenland Stadials) boundaries and GI-25 subdivision are indicated on the new AICC2023 chronology by horizontal bars.

Between 128 and 103 ka BP, the comparison between the AICC2012 timescale and the novel Dome Fuji ice core DF2021 chronology indicates that AICC2012 is likely too young by up to 4 kyr. Here, thanks to new highly resolved $\delta O_2/N_2$ data and to the alignment of $\delta^{18}O_{atm}$ and $\delta^{18}O_{calcite}$ records, we improve the consistency between AICC2023 and DF2021, now agreeing within 1.7 kyr over MIS 5e (Fig.12). With the new chronologies, the records of $\delta^{18}O_{atm}$ and $\delta O_2/N_2$ from Dome Fuji and EDC ice cores show synchronous variations between 140 and 115 ka BP although the $\delta O_2/N_2$ measurements from EDC are more scattered than DF data due to the use of smaller samples (see Supplementary material). However, $\delta D$ records still are slightly discordant and EDC record lags DF by up to 1,700 years over MIS 5.e and at the onset of the Antarctic Isotope Maximum (AIM) 24 (Fig. 12), suggesting some remaining chronology problems (AIM 24 onset) or regional climatic differences ($\delta D$ decrease over MIS 5.e). Between 180 and 150 ka BP, AICC2012 shows a better agreement with the DF2021 chronology than the new AICC2023 chronology which suggests younger ages as per TAC and $\delta^{18}O_{atm}$ dating constraints.



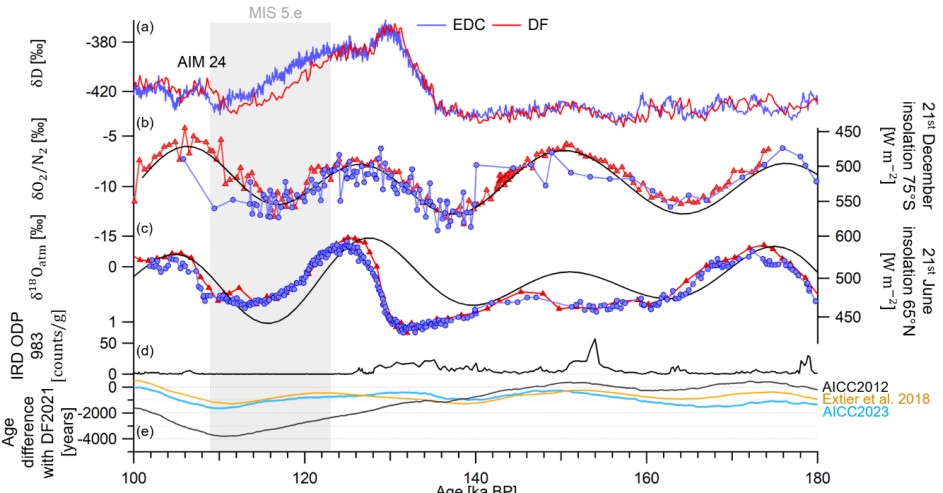

**Figure 12. Evolution of EDC and DF records on AICC2023 and DF2021 chronologies between 180 and 100 ka BP.** (a)
δD records from DF (red, Uemura et al., 2018) and EDC (blue, Jouzel et al., 2007). (b) $\delta O_2/N_2$ records from DF (red triangles,
Oyabu et al., 2022) and EDC (blue circles, this work). (c) $\delta^{18}O_{atm}$ records from DF (red triangles, Kawamura et al., 2007) and
EDC (blue circles, this work). DF and EDC records are represented on DF2021 and AICC2023 timescales. (d) IRD from ODP
983 (Barker et al., 2021). (e) Ice age difference between DF2021 and AICC2023 (blue), Extier et al. (2018) chronology (orange)
and AICC2012 (black). The age difference is calculated as per EDC age – DF2021 age. DF2021 age is transferred onto EDC
ice core via the volcanic synchronisation of Fujita et al. (2015). Grey rectangle indicates MIS 5e.

**4.2.2    MIS 11 (from 405 to 380 ka BP)**

Over the MIS 11 (from ~398 to 370 ka BP), the new AICC2023 chronology predicts older ages than
AICC2012 (by up to 2 kyr) with a diminished uncertainty (from 3.9 to 1.6 kyr). This shift towards older ages is
induced by $\delta^{18}O_{atm}$- $\delta^{18}O_{calcite}$ (Hulu, Sambao and Dongge caves) tie points at 377.3, 385.8 and 395.6 ka BP and
by the TAC age constraint at 362.1 ka BP (Fig. 8, 13). As a result, two major rises in the EDC atmospheric $CO_2$
and $CH_4$ concentration records (corresponding to Carbon Dioxide Jumps, CDJ+ 11a.3 and 11a.4, labelled as per
Nehrbass-Ahles et al., 2020) occur at 385.6 ± 1.4 and 389.8 ± 1.5 ka BP (Fig. 13). These two rapid jumps in $CO_2$
and $CH_4$ are better aligned with two abrupt decreases in the highly resolved $\delta^{18}O_{calcite}$ record of Zhao et al. (2019)
from Yongxing cave (dated at 386.4 ± 3.1 and 390.0 ± 3.0 ka BP) than when using the AICC2012 chronology
(improvement by ~800 years). Such millennial-scale synchronicity is expected between $CH_4$ and $\delta^{18}O_{calcite}$ series
from Chinese speleothems as they both are influenced by Asian monsoon area displacements (and associated
methane emissions from wetlands) (Sánchez Goñi et al., 2008).






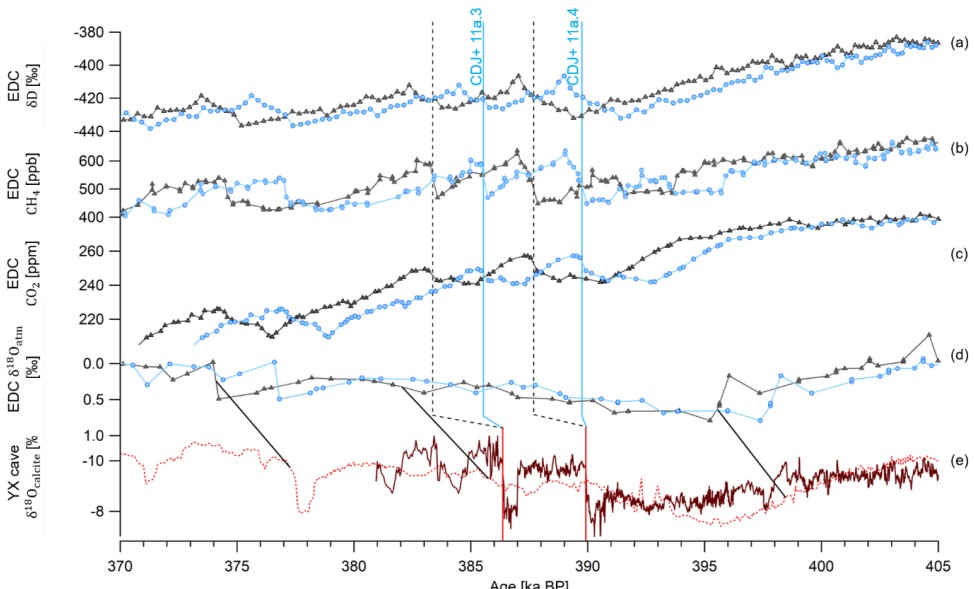

**779** **Figure 13. Evolution of climate tracers from EDC ice core and Yongxing cave stalagmites between 405 and 380 ka BP.**
**780** EDC records of (a) $\delta D$, (b) $CH_4$ (Nehrbass-Ahles et al., 2020), (c) $CO_2$ (Nehrbass-Ahles et al., 2020) and (d) $\delta^{18}O_{atm}$ on
**781** AICC2012 (grey triangles) and AICC2023 (blue circles) chronologies. (e) $\delta^{18}O_{calcite}$ from U-Th dated speleothems from Hulu,
**782** Dongge and Sambao cave (dashed red curve) and Yongxing (YX) cave (brown plain curve) (Zhao et al., 2019). CDJ+ are
**783** labelled as per Nehrbass-Ahles et al. (2020). Dashed black and blue vertical bars show jumps in $CO_2$ respectively on AICC2012
**784** and AICC2023 chronologies, red vertical bars show corresponding decreases in $\delta^{18}O_{calcite}$. Black lines show the three tie
**785** points between $\delta^{18}O_{atm}$ and SB cave $\delta^{18}O_{calcite}$ used to constrain AICC2023.

**786**

### 4.2.3 MIS 19 (from 780 to 760 ka BP)

**788** The Matuyama-Brunhes (MB) event (geomagnetic field reversal) is reflected by a globally synchronous
**789** event in the [10]Be signal: an abrupt termination of the large [10]Be peak following a long-term increasing trend
**790** recorded in both ice and sedimentary cores (Giaccio et al., 2023). The [40]Ar/[39]Ar age constrained chronology of a
**791** lacustrine succession from Sulmona basin (Giaccio et al., 2023) gives an age of $770.9 \pm 1.6$ ka BP for the [10]Be
**792** peak termination. The new AICC2023 chronology provides an estimate of $767.3 \pm 3$ ka BP for the same [10]Be peak
**793** termination, an age which is closer to the [40]Ar/[39]Ar age evaluation than the AICCC2012 chronology estimate
**794** ($766.2 \pm 3$ ka BP, Fig. 14). The new AICC2023 chronology indeed indicates an increasing older age than
**795** AICC2012 over MIS 19 (from 790 to 761 ka BP) due to the new $\delta^{18}O_{atm}$ based timescale (Fig. 8).

**796**

**797**

**798**

**799**



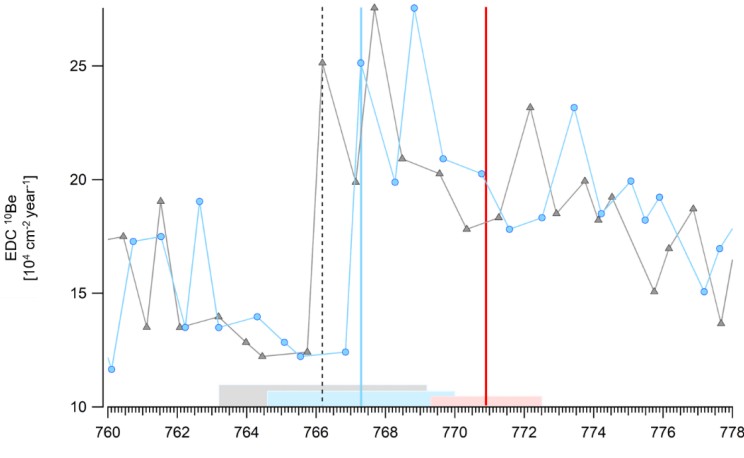

**Figure 14. EDC ¹⁰Be record on AICC2012 and AICC2023 chronologies between 778 and 760 ka BP.** Grey and blue vertical bars indicate the age of the abrupt EDC ¹⁰Be peak termination respectively on AICC2012 (grey triangles) and AICC2023 (blue circles) chronologies. The grey and blue horizontal squares correspond to AICC2012 and AICC2023 age uncertainties (±3 and ±2.7 ka respectively). The red vertical bar and horizontal square show the ¹⁰Be peak termination age and its error (Giaccio et al., 2023).

**Conclusions**

In this study, we have established a new reference chronology for EDC ice core, AICC2023 covering the last 800 kyr, that is consistent with the official GICC05 timescale over the last 60 kyr. A valuable update of the chronology construction has been the compilation of chronological and glaciological information including new age markers from recent high resolution measurements on the EDC ice core. As a result, the chronological uncertainty is reduced from 2.5 kyr in AICC2012 (standard deviation of 995 years) to 1.8 kyr on average here (standard deviation of 720 years). 90% of the new AICC2023 timescale is associated with an uncertainty lower than 2 kyr, against only 60% in the AICC2012 chronology. First, the distinct orbital chronologies derived from TAC, $\delta^{18}O_{atm}$ and $\delta O_2/N_2$ are coherent within their respective uncertainties except over three periods including MIS 11 and MIS 19. Second, new $\delta^{15}N$ measurements along with new sensitivity tests with the firn densification model described by Bréant et al. (2017) and adapted for the EDC ice core provide the most plausible evolution of LID at EDC over the last 800 kyr.

The majority of the age disparities observed between AICC2023 and AICC2012 chronologies are smaller than 2.3 kyr, hence minor considering the average uncertainty of AICC2012 (2.5 kyr). Exceptions are significant age shifts reaching 3.4, 3.8 and 5 kyr towards older ages respectively suggested over MIS 15, MIS 17 and MIS 19. However, most of these age discrepancies lead to an improved coherency between the new EDC timescale and independent absolute chronologies derived for other climate archives especially over the following periods: MIS 5, MIS 11 and MIS 19.



We have identified time intervals where building the chronology is more complicated such as TVI (from 540
to 456 ka BP) and from 800 to 600 ka BP, corresponding to the lowermost section of the core and we would like
to draw attention to the requirement for new measurements over these periods. In particular, the links between the
variability of TAC and $\delta O_2/N_2$ records and their orbital targets are not obvious over the 800 – 600 ka BP period
(Fig. 1). This may be due to bad quality of the ice and/or diffusion of gases through the ice matrix (Bereiter et al.,
2009). The imprecision of the signal may also be partially explained by the limited temporal resolution of the
existing dataset in this deep section. To address these issues, highly resolved TAC and $\delta O_2/N_2$ measurements are
needed in the lowermost section of EDC ice core. In addition, $\delta O_2/N_2$ from ice samples over the period covering
TVI should also be analyzed to investigate the mismatch between old and new datasets (Fig. 1).
A final important aspect would be to further extend the Paleochrono dating experiment by implementing other
ice cores such as Dome Fuji, WAIS (West Antarctic Ice Sheet) Divide and NEEM (North Greenland Eemian), for
which a large amount of chronological and glaciological information is now available.

**Author contribution**

Marie Bouchet wrote the manuscript with the contribution of all co-authors. Amaëlle Landais and Frédéric
Parrenin contributed to the conceptualization of the study and the methodology. Measurements on the EDC ice
core were performed at the LSCE by Antoine Grisart, Frédéric Prié, Roxanne Jacob and Elise Fourré. Emilie
Capron, Dominique Raynaud, Vladimir Ya Lipenkov and Marie-France Loutre contributed to the collection,
analysis and interpretation of the TAC record. Markus Leuenberger provided resources. The Krypton analysis was
conducted by Wei Jiang, Florian Ritterbusch, Zheng-Tian Lu, Guo-Min Yang. Thomas Extier, Anders Svensson,
Etienne Legrain and Patricia Martinerie contributed to the validation of the study.

**Competing interests**

At least one of the authors is a member of the editorial board of Climate of the Past. The peer-review process was
guided by an independent editor, and the authors have also no other competing interests to declare.

**Acknowledgements**

The research leading to these results has received funding from the European Research Council under the European
Union H2020 Programme (H2020/20192024)/ERC grant agreement no. 817493 (ERC ICORDA). Krypton
analysis has been supported by the Innovation Program for Quantum Science and Technology 2021ZD0303101,
and by the National Natural Science Foundation of China (41727901). Development of the Paleochrono model
was funded by CNRS/INSU/LEFE projects IceChrono and CO2Role. EC and EL acknowledge the financial
support from the French National Research Agency under the "Programme d'Investissements d'Avenir" (ANR-
19-MPGA-0001), through the Make Our Planet Great Again HOTCLIM project as well as the financial support
from the AXA Research Fund. We also acknowledge the assistance from the European Union FP5-EESD
Programme grant agreement no. EVK2-CT-2000-00077 (EPICA), the French National Research Agency
Programme "NEANDROOTS" (ANR-19-CE27-0011) and the French Polar Institute project no. 902
(GLACIOLOGIE CONCORDIA). Our special thanks go to Markus Grimmer, Marcel Haeberli, Daniel
Baggenstos, Jochen Schmitt, Matthias Baumgartner, Hubertus Fischer, Kenji Kawamura and Ikumi Oyabu for



sharing their thoughts and data, sustaining the discussion on the construction of new ice core age scales, and to
Sébastien Nomade and Alison Pereira for providing advice and expertise in geochronology.

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
