# Peer review of "The AICC2023 chronological framework and associated timescale"

_EGUsphere, 2023_

## Author Comment (AC2)

**1 Response to anonymous Referee #1**

We thank the reviewer for his/her valuable and helpful comments on the manuscript. We propose toimplement the following changes in a revised version.

4 Black = reviewer comment / blue = author's response / "*italic*" = revised text.

This paper presents an improved chronology for the Antarctic EPICA Dome C ice core for the time 5 6 interval 0-800 kyr. The development of this chronology involved various methods, including linking to 7 existing Greenland and Antarctic ice cores, orbital tuning with  $\delta^{18}O_{atm}$ ,  $\delta O_2/N_2$ , and total air content, and 8 employing firn modeling. One of the significant advancements is the improvement of a section around 110 ka BP, where several previous studies have pointed out that the AICC2012 chronology is too young. 9 Additionally, the increase of new gas data ( $\delta^{15}N$ ,  $\delta^{18}O$ ,  $\delta O_2/N_2$ , TAC) has greatly improved the precision 10 of orbital tuning and the estimation of the Lock-in depth scenario, reducing overall chronological 11 12 uncertainty significantly. While assessing whether the oldest part of the AICC2023 chronology has improved from the AICC2012 chronology is challenging, it does provide a reasonable estimate with a 13 14 larger age uncertainty compare to AICC2012.

- The paper is clearly written and convincingly demonstrates the method, including thorough sensitivity studies. The improved chronology for the EDC core is beneficial not only for the ice core community but also for the broader paleoclimate community. Therefore, I recommend accepting this paper for
- 18 publication in *Climate of the Past* after addressing the following comments.

**19 General comments:**

1) I am concerned about aligning the EDC  $\delta^{18}O_{atm}$  and the precession variations older than 590 ka 20 BP, although it seems to be a better solution than the previous one. While Extier et al. (2018) 21 suggested that the Heinrich-like events occurring especially during deglaciations delay the 22 response of  $\delta^{18}O_{atm}$  to orbital forcing, Oyabu et al. (2022) showed that the large lags of  $\delta^{18}O_{atm}$ 23 24 behind 65N summer insolation (~6 kyr) are not always seen during the Heinrich-like events. 25 For example, they showed that a large lag (>6 kyr) was found during the period of less IRD 26 (around the penultimate glacial maximum), while the lag for HE11 during Termination II is a 27 modest value of 4.1 kyr. Therefore, I think it would be valuable to indicate what potential errors may exist, although the authors have already given a safely large uncertainty. For example, what 28 about applying the same approach to well-dated periods such as the last glacial period, and/or 29 the range of time periods where  $\delta^{18}O_{atm}$ - $\delta^{18}O_{calcite}$  matching was conducted, with relatively small 30 dating uncertainties on speleothems, and comparing each other? This might serve as a test to 31 evaluate the reliability of the methodology, and the readers will be convinced of the reliability 32 33 of the obtained chronology.

34 Author's response: Thank you for these valuable inputs.

We will indicate the potential errors that may exist for using this approach for the period 590-800 ka BP and will refer to Oyabu et al. (2022). We agree with this limitation and this is the reason why we stick with a large uncertainty for the  $\delta^{18}O_{atm}$  tie points over this period (6 kyr).

38 We agree that such a comparison would be valuable in the Supplementary Material to support the use 39 of the approach presented in the manuscript. Following your suggestion, we tested three methodologies 40 to align  $\delta^{18}O_{atm}$  and precession over well-dated periods when  $\delta^{18}O_{atm}$ - $\delta^{18}O_{calcite}$  matching was done and 41 where we have high confidence in the chronology and we built three test chronologies:

42 1) The test chronology 1 is obtained by aligning  $\delta^{18}O_{atm}$  to 5-kyr-delayed precession as in Bazin et al. (2013).

- 44 2) The test chronology 2 is obtained by aligning  $\delta^{18}O_{atm}$  to precession as it would be expected if 45 only precession is driving the  $\delta^{18}O_{atm}$  signal through monsoon activity.
- 46 3) The test chronology 3 is obtained by aligning  $\delta^{18}O_{atm}$  to precession delayed if IRD counts are 47 superior to 10 counts/g and to precession without delay if IRD counts are inferior to 10 counts/g 48 (i.e., the same approach used in the paper for the period 590 – 800 ka BP when the  $\delta^{18}O_{atm}$ -49  $\delta^{18}O_{calcite}$  dating uncertainty becomes larger than 6 kyr and no East Asian speleothem  $\delta^{18}O_{calcite}$ 50 records are available before 640 ka BP).
- 51 We did an additional test to obtain a chronology derived from  $\delta^{18}O_{atm}$ - $\delta^{18}O_{calcite}$  matching only.
- 52 Over the 100-300 ka BP period, the test chronology 3 appears to be the best compromise as it agrees
- 53 well with both the AICC2023 age model and the chronology derived from  $\delta^{18}O_{atm}$ - $\delta^{18}O_{calcite}$  matching
- 54 (Figure). This is why we believe that it can be faithfully applied to the bottom part of the EDC ice core
- 55 while keeping large uncertainties in the tie points. These tests performed to justify our approach will be
- 56 implemented in the Supplementary Material.
- 57 This agreement is particularly satisfying over the 120-160 ka BP time interval which is the period on 58 which reviewer 2 wants a focus on. Over this period, Oyabu et al. (2022) identified a large peak (up to 59 61%) in the IRD record of McManus et al. (1999) (red plain line in panel e) and defined the associated HE 11 between 131 and 125 ka BP. Yet, if we consider the IRD record of Barker et al. (2021) used in 60 our study because it covers the last 800 kyr (blue plain line in panel e), we observe another large peak 61 (up to 56 counts/g) at around 150-156 ka BP. Because of this presence of IRD, to establish the test 62 63 chronology 3, we tuned  $\delta^{18}O_{atm}$  to the 5-kyr delayed precession over the whole period stretching from 155 to 124 ka BP (gray frame), which is larger than the duration covering only HE 11. The presence of 64 an IRD at 150-156 ka was not noted down in Oyabu et al. (2022) but is still visible in the McManus et 65 al. (1999) record displayed in their study (see Figure 11 of Oyabu et al., 2022). Because we have an IRD 66 at 150-156 ka, it justifies the lag observed in our chronology as well as in Oyabu et al. (2022) between 67  $\delta^{18}O_{atm}$  and precession. 68

Figure. EDC ice age difference between test chronology and AICC2023 between 300 and 100 ka BP. a) EDC ice age difference between AICC2023 and 4 tests chronologies: (i) test chronology 1 (grey dotted line), (ii) test chronology 2 (black dashed line), (iii) test chronology 3 (purple plain line) and (iv) test chronology derived using only δ18Oatm-δ18Ocalcite matching (red plain line). AICC2023 ice age 1σ uncertainty is shown by the red area. b)
δ18Oatm-δ18Ocalcite matching (red plain line). AICC2023 ice age 1σ uncertainty is shown by the red area. b)
δ18Oatm data from EDC (purple circles) and Vostok (blue circles). c) Precession delayed by 5 kyr (grey dotted line)
and not delayed (black dashed line) (Laskar et al. 2004). d) Derivative term of precession (black dashed line),
delayed precession (grey dotted line) and of the compiled δ18Oatm record (purple plain line). e) IRD (blue by Barker

et al. 2021; red by McManus et al. 1999). The gray squares indicate periods where IRD counts are superior to the
 10 counts/g threshold shown by the blue dotted horizontal line.

78 Regarding the gas age for the last 60 kyr, there are some age reversals in the AICC2012 chronology. I

believe that the AICC2023 chronology has improved as the tie points have been updated and there have
been significant progress on the construction of prior LIDs, but please make sure whether the AICC2023
chronology addressed and resolved the issue.

82 Author's response: The AICC2023 chronology resolved this issue. We believe this was due to the too 83 important variability of the analyzed LID scenario that is transferred to the  $\Delta$ depth, itself driven by the 84 high uncertainty associated with background LID. To address this problem, we revised the background 85 LID scenario using new data of  $\delta^{15}$ N and reduced its relative uncertainty to 10-20 % (where it was 86 evolving between 20 and 70% in the AICC2012 chronology). Here the less variable LID scenario results 87 in less gas age inversion. It can be seen in the figure below where EDC gas age augmentation rate is 88 plotted as a function of the age for AICC2012 (black) and AICC2023 (blue).

---

## Author Comment (AC3)

**Response to Anonymous Referee #2**

We thank the reviewer for his/her valuable and helpful comments on the manuscript. We propose to implement the following changes in a revised version.

Black = reviewer comment / blue = author's response / *"italic" = revised text.*

Review of Bouchet et al., 2023 – AICC2023

Bouchet et al., 2023 present an update to the AICC ten years after the first AICC. The update is focused on the EDC ice core and the older portion of the timescale that is based on orbital tuning. The increased density of measurements of d18Oatm, TAC, dO2/N2 and d15N are welcome and represent a significant improvement.

This manuscript describes a useful update the AICC. I remain confused by the exclusion of all(?) US, British, New Zealand, and Australian ice cores from the AICC. This is of relatively minor importance to this manuscript given that the different age ranges and the focus here on ages older than 100 ka and almost exclusively on EDC. I will urge that this chronology is names the EAICC – the East Antarctic Ice Core Chronology – given that there are more West Antarctic cores excluded from this chronology than East Antarctic cores that are included.

**Author's response:** Many thanks for this comment. We acknowledge that the aim of this study was perhaps not made very clear. The objective was indeed to focus on the long timescale (before 60 ka BP) to present the numerous new data available on the EDC ice core and using them to update AICC2012 with a special focus on deep time scales.

Adding the ice cores not yet included in the Paleochrono tool and mainly covering the last 60 kyr would be a different study which requests a lot of resources for the implementation of the ice cores in the Paleochrono tool. It was thus not possible to include everything in this study and we decided to focus more on the deep timescales with a particular focus on EDC. This will be better explained in the new manuscript.

As for the name of the chronology, after discussion with co-authors, we acknowledge that the name EAICC could have been a more suitable choice in the first place.

However, we would prefer to keep the name AICC for several reasons. It is less confusing and allows to show that AICC2023 is an update with respect to AICC2012 and that it should replace it. AICC provides an age model mainly for multiple glacial cycles where only the East Antarctic cores provide information. AICC uses age constraints from a set of cores (including NGRIP, so not just East Antarctic) and can be used as a template for West Antarctic cores as well (as for the Skytrain Ice Rise, Mulvaney et al. 2023). Finally, as mentioned above, one important future development would indeed be to include the high-resolution information from WAIS Divide and other cores.

The authors describe a large range of atmospheric gas measurements. The improvement in resolution of the many records is impressive. The orbital tuning of these records remains quite challenging and thus requires a myriad of subjective choices to develop both the timescale and the uncertainty. The orbital tuning, and the tuning to speleothem calcite, suffer from a lack of understanding in either cause of the variations in the measured parameter, the orbital parameter to tune to, or both. In particular, both O2/N2 and TAC have no process-driven explanation for why they vary based on the orbit characteristics and the variations are not produced by firn models. While this highlights the need for better understanding, particularly as great effort is going to extracting multiple >1Ma ice cores that reach the 40 ka world, it should not prevent doing the best that can be done with current understanding. And Bouchet et al. do this. They have produced a thoughtful chronology and while the manuscript is dense, it is also clearly written.

There are a couple of areas that stand out as areas of concern:

1) The firn modeling

This sentence is particularly confusing: "To obtain a coherent scenario, the firn modeling estimates have
been adjusted, by standard normalization, to the scale of LID values derived from $\delta15N$ data (later
referred to as experimental LID)."

This seems to hiding a major limitation in the methodology. If I understand correctly, the authors cannot
get the firn model to match the d15N-inferred firn thicknesses, so they just give up on the actual values
and instead seek to match the variations. Whether this is due to an inappropriate firn model (Breant) or
outdated forcing (the forcing isn't shown but I suspect the authors are using the classical isotope-
temperature scaling that Buizert et al. 2021 showed to be too cold at the LGM). The firn modeling should
really be done with multiple models – which is actually relatively easy to do thanks to the Community
Firn Model – and with a range of climate forcings. I think the authors efforts would be better served
employing other firn models and forcings rather than the impurity scenarios which the author reject.

**Author's response:** Thank you for raising this contradiction. The idea behind this proposition of fitting
the modeled LID (orange curve on Fig. 1) to experimental LID values was to avoid any discontinuity
when switching from experimental to modeled values when no data are available (grey rectangles on
Fig. 1).

[Figure]

**Figure 1.** Modeled LID and $\delta^{15}N$ data over the 0-3200 m depth interval. Grey rectangles indicate depth intervals
where $\delta^{15}N$ data are not available (either between 578 and 1086 m or between 1169 and 1386 m).

However, adjusting the modeled LID to experimental LID values induces a modification of 4.7 m at
most which remains within the background relative uncertainty (20%) so that the adjustment is small
and probably not really needed. This was already shown by the good comparison between modeled and
$\delta^{15}N$-inferred LID (L.91-140). To check this, we performed several Paleochrono runs to assess the
credibility of the two modeled LID scenarios (with and without adjustment).

On the depth interval from 578 to 1086 m, the raw background modeled scenario (orange curve, Fig. 2)
is almost as credible as the one that was adjusted (blue curve, Fig. 2) (i.e., close $\Delta_{no\ data}$ values). On the
second depth interval of interest, from 1169 to 1386 m, both scenarios show equal $\Delta_{no\ data}$ values, hence
equally credible.

[Figure]

**Figure 2.** Background and analyzed LID scenarios at EDC. a) Background LID as per AICC2023 (blue) and without fitting of the modeled LID to experimental LID values (orange). b) Analyzed LID. c) The averaged value of the misfit, $\Delta_{no\ data}$, is calculated for the two LID over the two depth intervals where $\delta^{15}N$ data are not available (either between 578 and 1086 m or between 1169 and 1386 m, see intervals shown by grey rectangles).

For more coherence, we believe that we should use the raw firn thickness predicted by the firn model, rather than fitting it to experimental LID values. This modification will be considered in the revised manuscript.

Finally, we would like to emphasize the facts that such modifications of the background LID scenario (less than 20% of the LID value) do not significantly affect the final age model and that the major improvement of the LID background scenario with respect to AICC2012 is the use of new highly-resolved $\delta^{15}N$ data over the 100-800 ka BP period.

We also noted the comment on the use of other firn models. Actually, we tested other firn models in a first instance (in particular the simple Herron and Langway model used also by Buizert et al., 2021) but we chose to keep the firn model outputs giving the best agreement with the $\delta^{15}N$ data over the last 800 kyr at EDC to fill the few gaps existing in the data series. The reason why we did not use the Buizert et al. (2021) approach is that it would require (i) a new EDC temperature scenario over the last 800 kyr while Buizert et al. (2021) only provided the temperature scenario over the last Termination as well as (ii) a new adapted temperature scenario for Vostok (which would be confusing for the readers since our goal is not to revise the Antarctic temperature reconstructions over the last climatic cycle). Indeed, to use the Buizert approach, we would have needed to adjust the temperature scenario so that the Herron and Langway model reproduces best the $\delta^{15}N$ data. We thus do think that testing the Breant model with different parameterizations (all of them published) and keeping the outputs resembling the most the $\delta^{15}N$-inferred LID was the simplest approach (and less confusing) to fill the few gaps in our $\delta^{15}N$-inferred LID.

2) I would like to see an analyses of the thinning function. The EDC AICC2012 thinning function does not decrease monotonically as expected from ice flow modeling (i.e. the input background scenario). If AICC2023 results in a smoother thinning function, this would provide significant support for the methodology.

**Author's response:** Although the new AICC2023 chronology reduces the absolute uncertainty of the
thinning function compared to AICC2012, it does not provide a smoother and strictly monotonous
scenario (see Figure).

However, we believe that this is not a problem for the following reasons:

(i) In a tube flow model, like Vostok's (Parrenin et al., 2004), the thinning function is not monotonous
since ice thickness variations are reflected in the thinning function. If the location of the dome at Dome
C shifted over the past 800 kyr, the same effect could have affected the thinning function.

(ii) There may also be non-laminar flow effects such as deformation due to more or less hard ice layers.

For instance, Dreyfus et al. (2007) described such a particularly complex thinning scenario at Dome C
over the MIS 15 (~580-560 ka BP).

[Figure]

**Figure**. Analyzed accumulation and thinning functions of EDC provided by AICC2012 and AICC2023 (black and
blue plain lines respectively) along with their absolute uncertainties (gray and yellow respectively). The
background thinning function is the same for AICC2012 and AICC2023 (dark blue dotted line).

General comments on Figure

For all figures, the timescale that each parameters is plotted on should be stated explicitly. It gets really
confusing when match points are connected with lines which are not vertical but the two parameters are
plotted on the same age x-axis.

**Author's response:** The changes will be made.

Vertical lines corresponding the major axes ticks would be really helpful in assessing the alignment of
features

**Author's response:** The changes will be made.

It would be really helpful to see the uncertainty assigned to each tie point. Presumably this could be
done with a horizontal bar on the match (on the EDC record)

**Author's response:** The changes will be made.

Specific comments

L36 – The introduction could really use subheadings.

**Author's response:** We agree and suggest the following subheadings:

1.1 Building age scales for deep polar ice cores
1.1.1 Motivation
1.1.2 Glaciological modeling
1.1.3 Chronological constraints derived from measurements
1.1.4 Bayesian dating tools
1.2 The AICC2012 chronology
1.3 The new AICC2023 chronology

L43 – "zipped" I don't think this is the right translation to English. I'm not sure what you are going for.
I think you are trying to say that a large amount of time is stored in a thin amount of ice.

**Author's response:** "zipped" will been changed to "stored".

L44 – need to make community possessive > community's

L44 – "core" not "cores"

L46 – add "the" before surface

**Author's response:** The changes will be made.

L53 – what about Nye?

**Author's response:** We suggest to change to: *"chronologies of ice cores at low-accumulation sites are*
*commonly established using ice flow and accumulation models (Nye, 1959; Schwander et al., 2001),*
*later on tied up with chronological and glaciological constraints (Bazin and Veres et al., 2013; Parrenin*
*et al., 2017)."*

L95 – I think it's worth emphasizing that Bender found no causal link between dO2/N2 and insolation
and was quite forthright about that.

**Author's response:** We believe that the quote from Bender (2002): *"We assert that insolation*
*influences snow metamorphism and grain properties in shallow firn. The insolation signature in these*
*properties is retained throughout the firn, and influences $O_2/N_2$ fractionation during bubble closeoff"* is
coherent with what we wrote in the introduction: *"observations led Bender (2002) to assert that local*
*summer solstice insolation affects near-surface snow metamorphism and that this imprint is preserved*
*as snow densifies in the firn and, later on, affects the ratio $\delta O_2/N_2$ measured in air bubbles formed at*
*the lock-in-zone."*

We agree to come back to this point if you believe it still needs modifications.

I don't expect that the authors will agree to incorporate WAIS Divide, but the introduction should have
a paragraph that acknowledges the exclusion and points readers to the timescales for these cores that are
tied to WAIS Divide as the best ones to use for past ~60 ka.

**Author's response:** We agree to designate the WD2014 chronology as the best candidate for the past
60 kyr. For greater coherence within the manuscript, we suggest to add a paragraph at the end of sect.
2.1 at L.206: *"We acknowledge the exclusion of the WAIS Divide ice core from the construction of the*
*AICC2023 age scale. Over the last 60 kyr, though, we recommend the use of timescales tied to the WAIS*
*Divide 2014 age model (WD2014, Buizert et al., 2015; Sigl et al., 2016). WD2014 hands over to*
*AICC2023 for the period older than 60 ka BP (that is for the section below the depth of 950 m for the*
*EDC ice core)."*

As mentioned above, the depth-depth correspondence between AICC2023 and WD2014 age models
will be given in supplement.

L118 – "peculiar" I think you mean "particular"

**Author's response:** We suggest to change *"peculiar"* to *"singular"* as "particular" is not exactly what
we meant.

L308 – shouldn't you reference Tison et al. 2015 here?

**Author's response:** The reference will be added.

More general comments

L349 – "superior" in English implies "better". I think "greater than" is better phrasing

**Author's response:** The change will be made.

L372 – the discarding of "tie points" worries me. Doesn't this imply that you don't understand the
underlying mechanisms that link the measurements parameter on the target tuning parameter? If you are
discarding tie points all together, should the uncertainty for the tie points you keep be increased to
respect that the relationship the ties are based on are not stationary?

**Author's response:** I think the word "discarding" was poorly chosen. Over the period of MIS 11 (gray
frame in the Figure), it is impossible to match $\delta O_2/N_2$ and insolation variations as they do not resemble
each other. For instance, two peaks in the insolation curve (dashed black line) only correspond to one
peak in the $\delta O_2/N_2$ data (blue circles). Hence, there is no tie point in the first place to be discarded. We
suggest the following modifications at lines 370-372: *"In such cases, the uncertainty associated with*
*each tie point is ranging from 6 to 10 kyr (precession half period) and some extrema in the target are*
*not used to tune the record (5 extrema over MIS 11 out of 63 over the last 800 kyr)."*

We argue that the uncertainty of 3 kyr for the $\delta O_2/N2$ tie points is enough. It was evaluated by Bazin et
al. (2016) who examined three $\delta O_2/N_2$ records from Vostok, Dome Fuji and EDC ice cores over MIS
5 and detected some site-specific $\delta O_2/N_2$ variations. This observation, along with the presence of a 100
kyr periodicity in the $\delta O_2/N_2$ record and the difficulty of identifying $\delta O_2/N_2$ mid-slopes and maxima,
led them to recommend the use of a 3 kyr uncertainty.

[Figure]

**Figure**. Alignment of $\delta O_2/N_2$ and insolation between 500 and 300 ka BP. (a) EDC raw $\delta O_2/N_2$ old data (black
circles for data of Extier et al. (2018) and purple squares for data of Landais et al. (2012)), outliers (grey crosses)
and filtered signal (black and purple lines). EDC raw $\delta O_2/N_2$ new data (blue triangles, this study) and filtered
signals (blue line). The $\delta O_2/N_2$ data are plotted on AICC2012 ice timescale. (b) Extrema in the compiled filtered
$\delta O_2/N_2$ dataset (blue plain line) are identified and matched to extrema in the (c) 21st December insolation at 75°
South plotted on a reversed y-axis and on the age scale given by Laskar et al. (2004) (dash line). The matching
peaks are linked by black vertical bars. (d) The 0 value in the time derivative of insolation (black line) and of the
filtered $\delta O_2/N_2$ dataset (blue line) corresponds to extreme values in the signals. The determined tie points between
$\delta O_2/N_2$ and insolation are depicted by markers on the horizontal line. Green circles are attached to a 3 kyr
uncertainty and purples squares are associated with a 6 kyr uncertainty (purple horizontal error-bar represented at
354.1 ka BP). Between 390 and 475 ka BP, all extrema are not tuned to the target due to the poor resemblance
between the signal and insolation (see gray frame).

L396 – The authors should not use "continuous" to describe the discrete gas measurements. These
samples are still quite sparse. Instead, the authors should emphasize increase in sample resolution and
the reduction in the largest gaps.

**Author's response:** We agree and several changes will be made to remove the adjectives "continuous"
or "discontinuous" when designating the gas records.

Figure 5 – I find the match points between d18O-O2 and speleothem d18O to be unconvincing. What
features are being matched and what features aren't seems arbitrary. Maybe this would be improved by
showing the uncertainty

**Author's response:** This point was also raised by the Referee 1 and we agree to modify the Fig. 5 so
that the uncertainty is shown.

L500 – I'm concerned the 6ka uncertainty is way to small. 6ka seems reasonable for the actual matches,
but shifting the tie points based by 5ka based on whether there is a Heinrich-like event is not well
founded. This really needs process modeling for support. Since that is outside the scope of the study, I
recommend increasing the uncertainty at least 10 ka (5ka since you don't know what to tune to and 5ka
for the murky matches themselves).

**Author's response:** Please don't forget that 6 ka is a 1 sigma uncertainty, so the 2 sigmas uncertainty
is 12 kyr, which seems enough for us.

Jouzel et al. (2002) presented the drawbacks of assuming a constant phase between $\delta^{18}O_{atm}$ and insolation which is a key assumption of the orbital tuning approach. To evaluate the uncertainty of the phasing between $\delta^{18}O_{atm}$ and insolation, Parrenin et al. (2001) assumed that the number of precessional cycles can be counted in the $\delta^{18}O_{atm}$ record. For them, this assumption *"is straightforward considering how clearly this cycle is imprinted in the $\delta^{18}O_{atm}$ series"* and implies that *"ice and gas chronologies are assigned to pass through a succession of large doors with a width of 6 kyr (1/4 of a precession cycle)"*. The authors estimated this width by combining glaciological modeling and orbital tuning.

We chose to stick with the recommendation of Jouzel et al (2002) and to use a 6-kyr uncertainty, which also allows to remain coherent with other orbital dating studies already conducted (Bazin et al., 2013). Yet, it is possible to run another dating experiment with the uncertainty increased to 10 kyr if needed.

L576 – As mentioned above, I don't really understand what you are doing to get a coherent scenario. Are there other firn models which get better agreement? And what are the climate forcings?

**Author's response:** We agree that the standard normalization is not necessary and are willing to let the rough modeled LID values.

L588 – Why are you not using the tie points to WAIS Divide directly? These ties are well established in Buizert et al. 2018. The WAIS Divide timescale is more accurate than GICC05 as demonstrated by Svensson et al. 2020 who had to shift the dates of GICC05 more than WDC14 for the bipolar matches.

**Author's response:** Although we agree, we would prefer to remain coherent with the AICC2012 study, that is to say to update the timescale AICC between 60 and 800 ka BP while keeping GICC05 between 60 and 0 ka BP. However, we understand fully this comment and will implement a correspondence between AICC2023 and WD2014 age models in the dataset submitted to PANGAEA. In the new version of the manuscript we will insist that for now, we focus mostly on the 60-800 ka BP age interval and stipulate that the WD2014 age model is more accurate over the last 60 kyr.

Ideally, one possible future development would indeed be to include the high-resolution information from WAIS Divide (and other cores). To do so, the WAIS Divide ice core should be added to the Paleochrono experiment along with the ties established by Buizert et al. 2018 and background glaciological scenarios that need to be determined. This development is beyond the scope of this study.

---

## Author Response (AR1)

**1 Final Response**

Black = reviewer or editor comment / blue = author's response / "grey" = former main text / "red" = revised main
 text.

**4 Summary:**

- 5 I/ Response to Referee 1 (L. 11)
- 6 II/ Response to Referee 2 (L. 531)
- 7 III/ Response to Editor (L. 814)
- 8 IV/ Additional comments (L. 947)
- 9 Line numbers refer to the revised versions of Supplementary Material and Main Text.
- 10

**11 I. Response to anonymous Referee #1**

We thank the reviewer for his/her valuable and helpful comments on the manuscript. We have implemented thefollowing changes in a revised version.

14 This paper presents an improved chronology for the Antarctic EPICA Dome C ice core for the time interval 0-800 kyr. The development of this chronology involved various methods, including linking to existing Greenland and 15 16 Antarctic ice cores, orbital tuning with  $\delta^{18}O_{atm}$ ,  $\delta O_2/N_2$ , and total air content, and employing firm modeling. One 17 of the significant advancements is the improvement of a section around 110 ka BP, where several previous studies 18 have pointed out that the AICC2012 chronology is too young. Additionally, the increase of new gas data ( $\delta^{15}$ N, 19  $\delta^{18}O, \delta O_2/N_2$ , TAC) has greatly improved the precision of orbital tuning and the estimation of the Lock-in depth 20 scenario, reducing overall chronological uncertainty significantly. While assessing whether the oldest part of the 21 AICC2023 chronology has improved from the AICC2012 chronology is challenging, it does provide a reasonable 22 estimate with a larger age uncertainty compare to AICC2012.

The paper is clearly written and convincingly demonstrates the method, including thorough sensitivity studies.
The improved chronology for the EDC core is beneficial not only for the ice core community but also for the broader paleoclimate community. Therefore, I recommend accepting this paper for publication in *Climate of the Past* after addressing the following comments.

**27 General comments:**

28 1) I am concerned about aligning the EDC  $\delta^{18}O_{atm}$  and the precession variations older than 590 ka BP, 29 although it seems to be a better solution than the previous one. While Extier et al. (2018) suggested that

- the Heinrich-like events occurring especially during deglaciations delay the response of  $\delta^{18}O_{atm}$  to orbital 30 forcing, Oyabu et al. (2022) showed that the large lags of  $\delta^{18}O_{atm}$  behind 65N summer insolation (~6 kyr) 31 32 are not always seen during the Heinrich-like events. For example, they showed that a large lag (>6 kyr) 33 was found during the period of less IRD (around the penultimate glacial maximum), while the lag for 34 HE11 during Termination II is a modest value of 4.1 kyr. Therefore, I think it would be valuable to indicate what potential errors may exist, although the authors have already given a safely large 35 uncertainty. For example, what about applying the same approach to well-dated periods such as the last 36 glacial period, and/or the range of time periods where  $\delta^{18}O_{atm}$ - $\delta^{18}O_{calcite}$  matching was conducted, with 37 relatively small dating uncertainties on speleothems, and comparing each other? This might serve as a 38 39 test to evaluate the reliability of the methodology, and the readers will be convinced of the reliability of 40 the obtained chronology.
- 41 Author's response: Thank you for these valuable inputs.

First, we modified the Sect. 3.2.3 (L. 503 in main text). We indicated the potential errors that may exist for usingthis approach for the period 590-800 ka BP and referred to Oyabu et al. (2022):

"Between 810 and 590 ka BP, the  $\delta^{18}O_{alm}$ ,  $\delta^{18}O_{calcite}$  dating uncertainty becomes larger than 6 kyr and no 44 East Asian speleothem  $\delta^{18}$ Ocalcite records are available before 640 ka BP. Over this time interval, we updated the 45 following approach of Bazin et al. (2013): EDC  $\delta^{18}O_{atm}$  and 5 kyr delayed climatic precession are synchronized 46 47 using mid-slopes of their variations. However, from the findings of Extier et al. (2018),  $\delta^{18}O_{atm}$  should rather be aligned to precession without delay when no Heinrich-like events occurs. Indeed,  $\delta^{18}O_{atm}$  is sensitive to both orbital 48 49 and millennial scale variations of the low latitude water cycle (Capron et al., 2012; Landais et al., 2010) and 50 Heinrich-like events occurring during deglaciations delay the response of  $\delta^{18}O_{atm}$  to orbital forcing through 51 southward ITCZ shifts (Extier et al., 2018). We thus chose to align  $\delta^{18}$ Oatm to precession when no Ice Rafted Debris 52 (IRD) peak is visible on the studied period in the ODP983 record (Barker, 2021) and keep a 5 kyr delay when IRD 53 peaks are identified. This results in shifting 12 tie points of Bazin et al. (2013) by 5,000 years towards older ages 54 (see Fig. 6). The eight remaining tie points of Bazin et al. (2013) that coincide with peaks in the IRD record are 55 kept (Fig. 6). To confirm the validity of our approach, we tested three methodologies to align  $\delta^{18}O_{atm}$  and precession over well-dated periods when  $\delta^{18}O_{atm}$ -  $\delta^{18}O_{calcite}$  matching was done (see Sect. 2.2.2 in the 56 57 Supplementary Material). These tests support our approach but in order to account for potential errors associated 58 with this tuning method (Oyabu et al., 2022), a 6 kyr uncertainty (1 $\sigma$ ) is attributed to the  $\delta^{18}O_{atm}$  derived tie points 59 over the period between 810 and 590 ka BP.

60 Figure 5. Alignment of EDC  $\delta^{18}O_{atm}$  and climatic precession between 810 and 590 ka BP. (a) Compiled EDC

61  $\delta^{18}O_{atm}$  on AICC2012 gas timescale. (b) Precession delayed by 5,000 years (grey dashed line) and not delayed 62 (black dashed line) (Laskar et al., 2004). (c) Temporal derivative of precession (black dashed line), delayed

63 precession (grey dotted line) and of the compiled  $\delta^{18}O_{atm}$  record (purple plain line). (d) Ice-Rafted Debris at

64 ODP983 site (North Atlantic Ocean, southwest of Iceland) by Barker (2021). The gray squares indicate periods

where IRD counts are superior to the 10 counts/g threshold shown by the blue dotted horizontal line. Grey vertical bars illustrate new tie points between EDC  $\delta^{18}O_{atm}$  and delayed precession mid-slopes (i.e. derivative extrema)

bars illustrate new tie points between EDC  $\delta^{18}O_{atm}$  and delayed precession mid-slopes (i.e. derivative extrema) when IRD counts are superior to the threshold. Black vertical bars illustrate new tie points between EDC  $\delta^{18}O_{atm}$

68 and precession mid-slopes (i.e. derivative extrema) when no Heinrich-like events is shown by IRD record. The 12

69 kyr  $2\sigma$ -uncertainty attached to the tie points is shown by the horizontal error-bars in panel b."

Then, we agree that applying the same approach over well-dated periods where  $\delta^{18}O_{atm}-\delta^{18}O_{calcite}$  matching was conducted would be valuable in the Supplementary Material to support the use of the approach presented in the manuscript. Following your suggestion, we modified the Sect. 2.2 at L.59 in the Supplementary Material as follows:

**74 2.2 "Aligning EDC $\delta^{18}O_{atm}$ record and climatic precession variations**

For the construction of the new AICC2023 chronology between 800 and 590 ka BP, the EDC  $\delta^{18}O_{atm}$  record is

76 aligned with the climatic precession delayed or not by 5,000 years depending on the occurrence of Heinrich like

- events, reflected by peaks in the IRD record from the North Atlantic Ocean (Sect 3.2.3 in the main text). Potential
- errors may arise from aligning  $\delta^{18}O_{atm}$  to precession (Oyabu et al., 2022). To support the use of our approach, we
- 79 test three methodologies to align  $\delta^{18}O_{atm}$  and precession. Four test chronologies are built:

| 80 | 1)  | The test chronology 1 is obtained by aligning $\delta^{18}O_{atm}$ to 5-kyr-delayed precession as in Bazin et al. (2013).     |
|----|-----|-------------------------------------------------------------------------------------------------------------------------------|
| 81 | 2)  | The test chronology 2 is obtained by aligning $\delta^{18}O_{atm}$ to precession as it would be expected if only precession   |
| 82 |     | is driving the $\delta^{18}O_{atm}$ signal.                                                                                   |
| 83 | 3)  | The test chronology 3 is obtained by aligning $\delta^{18}O_{atm}$ to precession delayed if IRD counts are superior to 10     |
| 84 |     | counts/g and to precession without delay if IRD counts are inferior to 10 counts/g.                                           |
| 85 | 4)  | The test chronology 4 is obtained by matching $\delta^{18}O_{atm}$ and $\delta^{18}O_{calcite}$ variations only.              |
| 86 |     |                                                                                                                               |
| 87 | 2.2 | .1 Between 810 and 590 ka BP                                                                                                  |
| 88 | We  | first evaluate the impact on the chronology whether $\delta^{18}O_{atm}$ is aligned with the precession with or without delay |
| 89 | bet | ween 810 and 590 ka BP. The age mismatch between test chronologies 1 and 2 is of 3,000 years on average,                      |

- 90 reaching its maximum value of 3,700 years at  $712 \pm 2.6$  ka BP (red arrow in Fig. S4).